# Aryl hydrocarbon receptor utilises cellular zinc signals to maintain the gut epithelial barrier

Xiuchuan (Lucas) Hu[1,2,10], Wenfeng Xiao[1,3,10], Yuxian Lei[4], Adam Green[2], Xinyi Lee[2], Muralidhara Rao Maradana[5], Yajing Gao[1,3], Xueru Xie[1,3], Rui Wang[2], George Chennell[6], M. Albert Basson[7,8], Pete Kille[9], Wolfgang Maret[2], Gavin A. Bewick[4], Yufeng Zhou[1,3] ✉ & Christer Hogstrand[2] ✉

Zinc and plant-derived ligands of the aryl hydrocarbon receptor (AHR) are dietary components affecting intestinal epithelial barrier function. Here, we explore whether zinc and the AHR pathway are linked. We show that dietary supplementation with an AHR pre-ligand offers protection against inflammatory bowel disease in a mouse model while protection fails in mice lacking AHR in the intestinal epithelium. AHR agonist treatment is also ineffective in mice fed zinc depleted diet. In human ileum organoids and Caco-2 cells, AHR activation increases total cellular zinc and cytosolic free $Zn^{2+}$ concentrations through transcription of genes for zinc importers. Tight junction proteins are upregulated through zinc inhibition of nuclear factor kappa-light-chain-enhancer and calpain activity. Our data show that AHR activation by plant-derived dietary ligands improves gut barrier function at least partly via zinc-dependent cellular pathways, suggesting that combined dietary supplementation with AHR ligands and zinc might be effective in preventing inflammatory gut disorders.

The small intestinal epithelium is a single cell-layer composed of many repetitive and self-renewable crypt-villus units[1–3]. Its main functions include allowing the selective passage of nutrients into the body whilst separating the intestinal epithelium from the luminal contents, thereby protecting against pathogenic bacteria, antigens, and other harmful agents[4–7]. The intestinal barrier function is governed by paracellular tight junctions of the epithelium, the mucus layer, and the enteric immune system[8]. The barrier function is compromised in IBD conditions, such as Crohn's disease and ulcerative colitis, causing leakage and inflammation[6,9]. Conversely, inflammation of the enteric immune system may cause dysregulation of the epithelial barrier and IBD.

Both the aryl hydrocarbon receptor (AHR) and zinc are essential for the development of a fully differentiated intestinal epithelium and maintenance of its integrity and are also involved in regulation of both the innate and adaptive immune responses[10–16]. In zinc deficiency, growth of intestinal stem cells (ISCs) is attenuated, and tight junctions are compromised, resulting in increased epithelial leakiness[17–19]. This

[1]Institute of Pediatrics, Children's Hospital of Fudan University, and the Shanghai Key Laboratory of Medical Epigenetics, International Co-laboratory of Medical Epigenetics and Metabolism, Ministry of Science and Technology, Institutes of Biomedical Sciences, Fudan University, Shanghai, China. [2]Department of Nutritional Sciences, King's College London, London, UK. [3]National Health Commission (NHC) Key Laboratory of Neonatal Diseases, Fudan University, Shanghai, China. [4]Department of Diabetes, Cardiovascular and Metabolic Medicine & Sciences, Faculty of Life Science and Medicine, King's College London, London, UK. [5]The Francis Crick Institute, London, UK. [6]Clinical Neuroscience Department, King's College London, London, UK. [7]Centre for Craniofacial and Regenerative Biology and MRC Centre for Neurodevelopmental Disorders, King's College London, London, UK. [8]Clinical and Biomedical Sciences, University of Exeter Medical School, Exeter, UK. [9]School of Biosciences, Cardiff University, Cardiff, UK. [10]These authors contributed equally: Xiuchuan (Lucas) Hu, Wenfeng Xiao. ✉e-mail: yfzhou1@fudan.edu.cn; christer.hogstrand@kcl.ac.uk

may explain why zinc supplementation is widely used to improve intestinal epithelial barrier function and prevent diarrhoea in children as well as in farm animals[12,20]. The AHR is a transcription factor belonging to the basic helix-loop-helix/Per-ARNT-SIM (bHLH-PAS) protein family[21]. Nutritionally relevant agonists of AHR include tryptophan metabolites and indoles mainly found in cruciferous vegetables, such as broccoli and cabbage[22]. Recent research revealed that AHR activation by plant-based dietary ligands in the intestine regulate intestinal stem cell (ISC) proliferation and differentiation, and epithelial barrier function[14,23,24]. Mice with defective AHR signalling in the intestinal epithelium have an increased susceptibility to enteric infection, which could be corrected by dietary supplementation with AHR ligands[25]. Whether there is a relationship between AHR and zinc in maintaining the gut epithelial barrier function is unknown, but a recent study shows that body stores in *Drosophila melanogaster* is dependent on dietary intake of tryptophan, which is metabolised to kynurenine[26], a potent AHR agonist[27]. In the present study we explore the hypothesis that AHR regulates zinc uptake into the intestinal epithelium resulting in an improved barrier function.

## Results

### The protective effect of I3C in dextran sodium sulfate-induced IBD juvenile mouse model depends on dietary zinc intake

Since both AHR activation and zinc improve intestinal barrier function, we hypothesised that the protective effect of AHR agonists on gut inflammation is dependent on the dietary zinc supply. To test our hypothesis, we used a dextran sodium sulfate (DSS) induced IBD juvenile mouse model and treated the animals by daily gavage with a plant-derived AHR ligand precursor, indole-3-carbinol (I3C), in combination with zinc-depleted (5 mg/kg), zinc-replete (35 mg/kg), or zinc-supplemented (100 mg/kg) feed as outlined diagrammatically in Fig. 1a. Neither the different zinc diets nor I3C gavage had an effect on body weight without DSS treatment (Supplementary Fig. 1c). DSS treatment caused pronounced body weight loss (Supplementary Fig. 1c), diarrhoea and blood in the faeces (Fig. 1b) with the most severe effects observed in mice fed the zinc-depleted diet. I3C treatment significantly alleviated the body weight loss and intestinal histopathology in DSS-induced IBD for mice on 100 or 35 mg/kg zinc diets, but not in the 5 mg/kg zinc diet group. Furthermore, reduced colon length, a marker of intestinal inflammation with very low variability in the DSS-induced IBD model, was less severe in mice treated with I3C in combination with diets containing 35 or 100 mg/kg zinc; however, DSS-induced decrease in colon length could not be rescued by I3C in mice fed on the 5 mg/kg zinc diet group (Fig. 1c). DSS-treated mice developed multiple erosive lesions and significant inflammatory responses in colon and ileum, characterised by increased inflammatory cell infiltrations, goblet cell loss, crypt abscess formation and submucosal oedema. I3C-treated mice fed on diets containing 35 or 100 mg/kg zinc displayed lower levels of inflammatory lesions compared with those on the same diets without I3C treatment (Fig. 1d and Supplementary Fig. 1d) and this was found to be statistically significant when quantified using blinded scoring (Fig. 1e and Supplementary Fig. 1e). However, no benefit of I3C treatment was observed in DSS-exposed mice given the 5 mg/kg zinc diet. Gut barrier integrity is in part maintained by tight junction proteins such as occludin (OCLN) and the mucosal barrier, which is modulated by mucin-2 (MUC2) expression. Expression of MUC2 has been shown to be reduced in an in vitro goblet cell model (HT-29-MTX) following treatment with zinc depleted media[28] and zinc was found to simulate secretion of mucus in the fish intestine[29]. Therefore, to evaluate intestinal barrier function, we determined expression levels of OCLN and MUC2 in colon tissues by immunohistochemistry (IHC). The IHC results showed abundant OCLN and MUC2 in the colonic epithelial cells in all conditions without DSS, while expression of these proteins was

lower in mice given DSS (Fig. 1f and Supplementary Fig. 2a, b). I3C treatment of mice exposed to DSS resulted in higher expression of OCLN and MUC2 than mice not treated with I3C, but only in the groups fed on the 35 or 100 mg/kg zinc diets (Fig. 1f). Mice fed on the 5 mg/kg zinc diet did not respond to I3C in terms of maintaining higher OCLN and MUC2 expression following DSS exposure. RT-qPCR analysis of inflammatory cytokines and genes in the colon, such as *Il6, Tnf-α, Cox2, Nos2* and inflammatory chemokine *Ccl2*, showed that groups fed on 35 or 100 mg/kg zinc diets with I3C treatment presented a less inflammatory state compared to groups feed on 5 mg/kg zinc diets with or without I3C treatment (Supplementary Fig. 1f). Together, our results indicate that the protective effect of I3C in DSS-induced IBD model is dependent on sufficient dietary zinc intake.

### The protective effect of I3C and dietary zinc intake in the DSS-induced IBD juvenile mouse model depend on AHR in intestinal epithelial cells

To verify if the beneficial effect of I3C against DSS-induced colitis depended on AHR expression in intestinal epithelial cells, we generated intestinal epithelial-specific AHR knockout mice (villin^cre *Ahr*^fl/fl) and repeated our DSS experiments (Fig. 2a). As predicted, I3C did not mitigate against DSS induced IBD; disease activity (Fig. 2b), weight loss (Supplementary Fig. 3a), colon length (Fig. 2c), and histopathology and inflammatory state of the colon (Fig. 2d, e and Supplementary Fig. 3e) and ileum (Supplementary Fig. 3b, c) were not different between groups. Remarkably, zinc supplementation alone did not reduce IBD in AHR deficient mice. These results show that intestinal AHR activation by orally administered I3C ameliorates IBD in the DSS mouse model, but the efficacy of the treatment is dependent on a zinc-sufficient diet and is most effective together with zinc supplementation.

### 16S rDNA sequencing analysis of the intestinal content from WT and villin^cre*Ahr*^fl/fl mice in DSS-induced IBD juvenile mouse model

We considered that some of the effects of I3C and zinc treatment observed might have been mediated by influence of these treatments on gut microbiota. We therefore performed 16S rDNA sequencing of the intestinal content of four mice from each DSS group. Dietary zinc concentrations and I3C administration did not affect alpha or beta diversity of species in WT mice, but there was a small increase in alpha diversity of microbes with increasing dietary zinc levels in DSS treated villin^cre*Ahr*^fl/fl mice (Supplementary Fig. 4a, b). There was also a higher alpha diversity in villin^cre*Ahr*^fl/fl mice compared with the WT, independent of zinc concentration in the diet and this could be attributed to an increase in abundance of several pathogenic species (Supplementary Fig. 4c–e). We conclude from this analysis that changes in microbiota cannot explain the benefit of zinc and I3C co-treatment on mitigation of DSS-induced pathology.

### 6-Formylindolo(3,2-b)carbazole (FICZ) and zinc treatment improve barrier function of human Caco-2 cells and ileum organoids

Next, we used in vitro models to explore the relevance of our findings in human. We used differentiated Caco-2 cells grown into 2D epithelia which differentiate into enterocytes with characteristics similar to those of small intestine enterocytes[30], and 3D cultures of human ileum organoids derived from healthy donors. Since I3C is a pro-ligand of AHR and does not function without conversion in the gut, we used 6-formylindolo(3, 2-b)carbazole (FICZ) as the AHR agonist in our in vitro experiments. To test if AHR activation requires zinc to promote epithelial resistance and reduce paracellular permeability we grew Caco-2 cells on inserts in a Transwell® system. With this system we could mimic the barrier function of the epithelium of the human small intestine and measured its trans-epithelial electrical resistance (TEER)

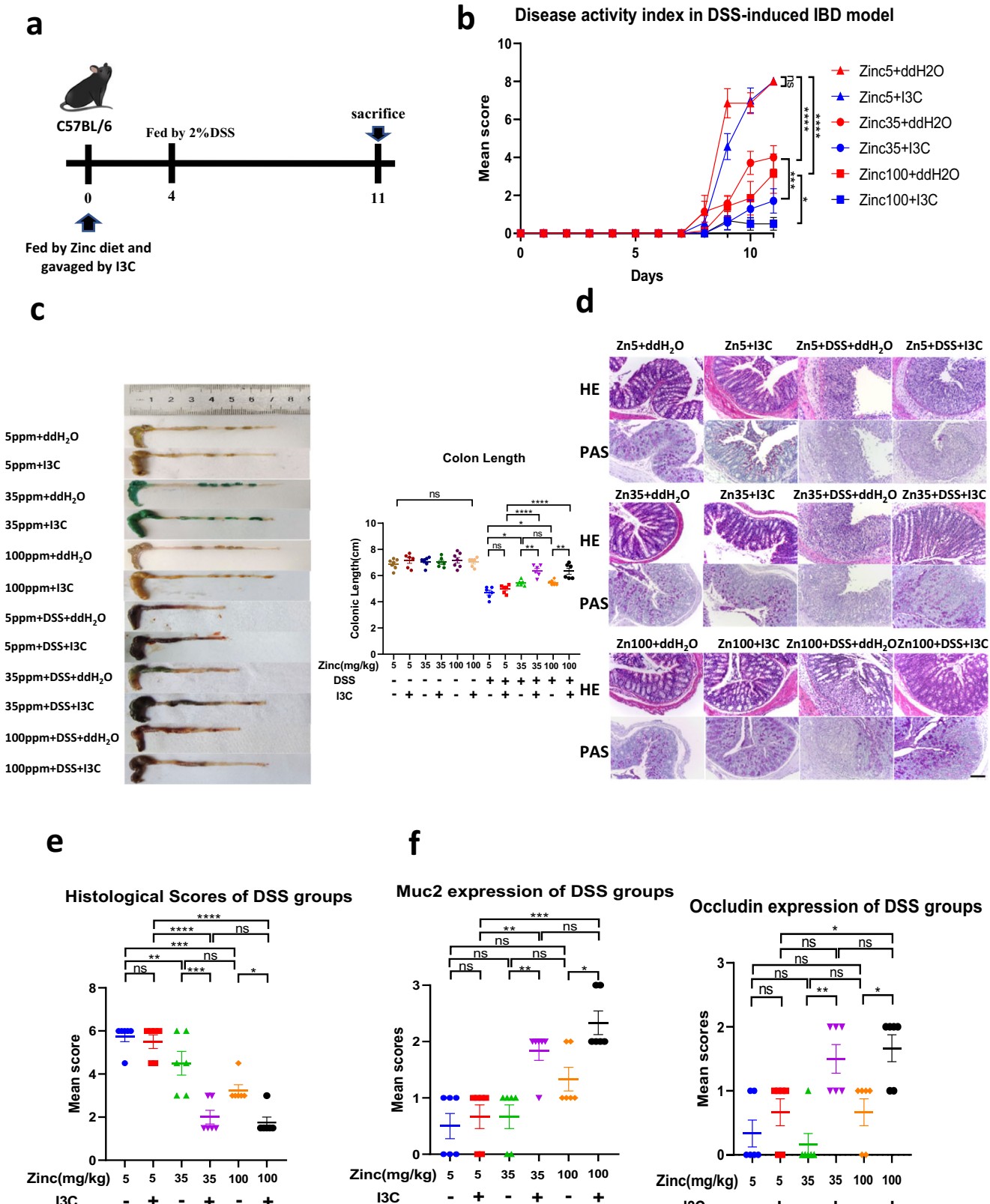

and permeability (leakage of FITC-Dextran 4000) employing different experimental conditions to simulate a healthy gut, inflammation (DSS treatment), and hypoxia. Addition of zinc or FICZ alone increased electrical resistance and reduced permeability in all experimental conditions but the effect was greater when both agents were added together (Fig. 3a and Supplementary Fig. 5a–e). The benefit of the combined treatment with zinc and FICZ on epithelial tightness was particularly strong in the two models of compromised epithelia through exposure to DSS (Fig. 3a and Supplementary Fig. 5d) or hypoxia (Supplementary Fig. 5b, e) although it was also significant in

**Fig. 1 | Effects of zinc deficiency and I3C treatment on intestinal integrity in the DSS-induced IBD mouse model. a** Schematic representation of the experimental schedule. Three-week-old C57BL/6 J mice were provided with diets containing one of three zinc concentrations (5 mg/kg (Zinc5), 35 mg/kg (Zinc35) and 100 mg/kg (Zinc100) from Days 0 to 10 with (I3C) or without (ddH$_2$O) I3C given by daily gavage. DSS was administered by the drinking water from Day 4 to Day 10. Mice were sacrificed on Day 11. **b** Changes in intestinal disease activity index, based on diarrhoea and bleeding. Each data point represents the mean for seven animals ($n = 7$). **c** Colon image (left) and colon lengths (right) measured on Day 11 ($n = 6$ animals). **d** Histopathological changes in the colon tissue were examined by H&E and Periodic Acid Schiff[58] staining (magnification, ×100). Scale bar, 100 μm ($n = 6$ animals). **e** Histopathological scores of the colon tissue in DSS treated groups ($n = 6$ animals). **f** Immunohistochemistry scoring of MUC2 and Occludin protein expression in colon tissue ($n = 6$ animals). Animal experiments were repeated twice. Data are presented as mean values and error bars show SEM. Statistical analysis was performed using 1-way ANOVA followed by Tukey's multiple comparison tests. $*p < 0.05$, $**p < 0.01$, $***p < 0.001$, $****p < 0.0001$, ns not significant.

unchallenged cells representing a healthy intestinal epithelium (Supplementary Fig. 5a, c). Human ileum organoids were used to investigate the protective properties of FICZ and zinc on intestinal epithelium barrier function in a system with closer resemblance to the human intestine. Treatment of organoids with 60 μM EDTA for 24 h to challenge epithelial permeability of organoids was effective in increasing leakage of FITC-Dextran 4000 into the organoid lumen (Supplementary Fig. 5f). Also in this organoid system, the combined treatment of zinc and FICZ reduced epithelial permeability in a seemingly additive way (Fig. 3b and Supplementary Fig. 5g). Thus, treatment with zinc and FICZ in combination markedly improves barrier function of human ileum organoids and Caco-2 cells.

### FICZ and zinc treatment improve expression of TJ proteins in human Caco-2 cells and ileum organoids

Next, to examine if the improved barrier function was related to increased expression of genes for TJ proteins, we treated Caco-2 cells grown to a differentiated epithelium with 8 μM zinc and 100 nM FICZ either separately or in combination and analysed abundance of mRNA for zona occludens-1 (ZO-1), occludin (OCLN) and claudin-3 (CLD-3). Both 8 μM zinc or 100 nM FICZ independently increased expression of *ZO-1*, *OCLN* and *CLD-3* in Caco-2 cells, with a stronger effect when given as combination treatment (Fig. 3c). Chelation of the 8 μM zinc added to the medium by an equimolar concentration of the cell permeant zinc chelator, tris(2-pyridylmethyl)amine (TPA)[31], completely abolished the effects of zinc and FICZ on expression of *ZO-1* and *CLD-3* while diminishing those on expression of *OCLN* (Fig. 3c). Overwhelming the chelating capacity of TPA by addition of another 8 μM zinc (16 μM in total) restored the effect of zinc and FICZ on *ZO-1*, *OCLN*, and *CLD-3* gene expression showing that the effects of FICZ and changing the zinc concentration of the medium on genes for TJ proteins were dependent on intracellular zinc. The effects of zinc and FICZ on TJ gene expression were confirmed in human ileum organoids treated with 0, 25 or 50 μM zinc added to the medium in presence of 10 μM Ca-EGTA (to reduce background zinc in the medium) with or without FICZ (Fig. 3d). In addition, we found that expression of mucin-2 (MUC2) mRNA was increased in the human ileum organoids treated with zinc and FICZ (Fig. 3d), supporting the data from the mouse experiment (Figs. 1c and 2d and Supplementary Figs. 1c and 3d, f) and previous findings in an in vitro goblet cell model[28]. Western blot analysis of ZO-1 and OCLN abundance in Caco-2 cells indicated that expression of both TJ proteins increased after treatment with either zinc or FICZ, but treatment of cells with the combination of zinc and FICZ had a stronger effect (Fig. 3e). Again, the effect of zinc addition to the medium on TJ protein expression appeared to be caused primarily by an increase in intracellular zinc concentration as shown by intracellular zinc chelation using TPA (Fig. 3e). The effects of zinc and FICZ on OCLN and MUC2 were identified and quantified by immunocytochemistry in human ileum organoids (Fig. 3f). These results demonstrated that treatment of human intestinal epithelial cells with zinc and FICZ induced expression of mRNA for TJ proteins and *MUC2*, and this translated into increased abundance of the respective proteins, providing a plausible explanation for the enhanced barrier function following the same treatment.

### FICZ and zinc treatment inhibits the activities of NF-κß and calpain to promote the expression of tight junctions in human Caco-2 cells and ileum organoids

We subsequently investigated the mechanisms by which zinc and AHR ligands might enhance abundance of TJ proteins. The NF-κß pathway has been shown to transcriptionally suppress genes for TJ proteins through NF-κß P65 binding to their promoters[32]. AHR activation has proven effective in attenuating NF-κß activation, mainly through interfering with P65 recruitment to DNA[33]. Zinc can inhibit NF-κß signalling by preventing phosphorylation of P65 and Iκßα[34]. We hypothesised that zinc and AHR signalling may converge on and effectively inhibit activation of the NF-κß pathway. We therefore investigated the effects of 8 μM zinc and 100 nM FICZ individually or in combination on the NF-κß pathway in Caco-2 cells treated with tumour necrosis factor alpha (TNF-α) to stimulate activating serine-536 phosphorylation of P65 and serine-32 phosphorylation of Iκßα. Addition of either zinc or FICZ to the medium attenuated TNF-α induced phosphorylation of P65 and Iκßα and the combination treatment was the most potent in both cases (Fig. 4a). We therefore repeated the measurement of expression of genes encoding TJ proteins in Caco-2 cells after treatment with zinc and/or FICZ, but this time in the presence of an NF-κß blocker, EVP4593 (Supplementary Fig. 6a). In the presence of 10 μM EVP4593, zinc and/or FICZ had no stimulatory effect on expression of *ZO-1* and *OCLN*, indicating that zinc and FICZ increase expression of mRNA for TJ proteins by alleviating transcriptional inhibition by NF-κß.

TJ proteins are degraded by calpain proteases, which are inhibited by zinc[35,36]. We therefore investigated the effects of 8 μM Zinc and 100 nM FICZ individually or in combination on calpain activity in Caco-2 cells grown to an epithelium and in human ileum organoids. Addition of either zinc or FICZ to the medium reduced calpain activity in Caco-2 cells and human ileum organoids, and the combination treatment was the most potent in inhibiting calpain (Fig. 4b, c). We therefore confirmed the expression on TJ proteins after zinc and/or FICZ treatment in Caco-2 cells grown to a differentiated epithelium, but at this time in the presence of a calpain inhibitor, calpeptin (Supplementary Fig. 6b). In the presence of 50 μM calpeptin, zinc and/or FICZ had no stimulatory effect on abundance of TJ proteins, suggesting that inhibition of calpain is a second mechanism by which zinc and FICZ improve Caco-2 cell epithelium barrier function.

### AHR promotes zinc uptake in cells and organoids by transcriptional activation of zinc importers

Since AHR activation by FICZ influenced expression of TJ genes and proteins in a zinc-dependent manner, we tested if AHR activation by FICZ induces transcription of genes for zinc regulatory proteins. We previously published, based on chromatin immunoprecipitation microarray (ChIP-chip) mouse data obtained from Dere et al.[37], that AHR has functional binding sites in several zinc-regulatory genes[38]. Analysis of a more recent ChIP-seq dataset[39] from human MCF-7 cells exposed to the xenobiotic AHR agonist, 2,3,7,8-tetrachlorodibenzo-p-dioxin (TCDD), confirmed AHR binding to numerous zinc-regulatory genes, including genes for several zinc importers of the ZIP (SLC39) family. To determine if FICZ induces expression of the zinc-regulatory genes identified as potentially being trans-activated by AHR, we

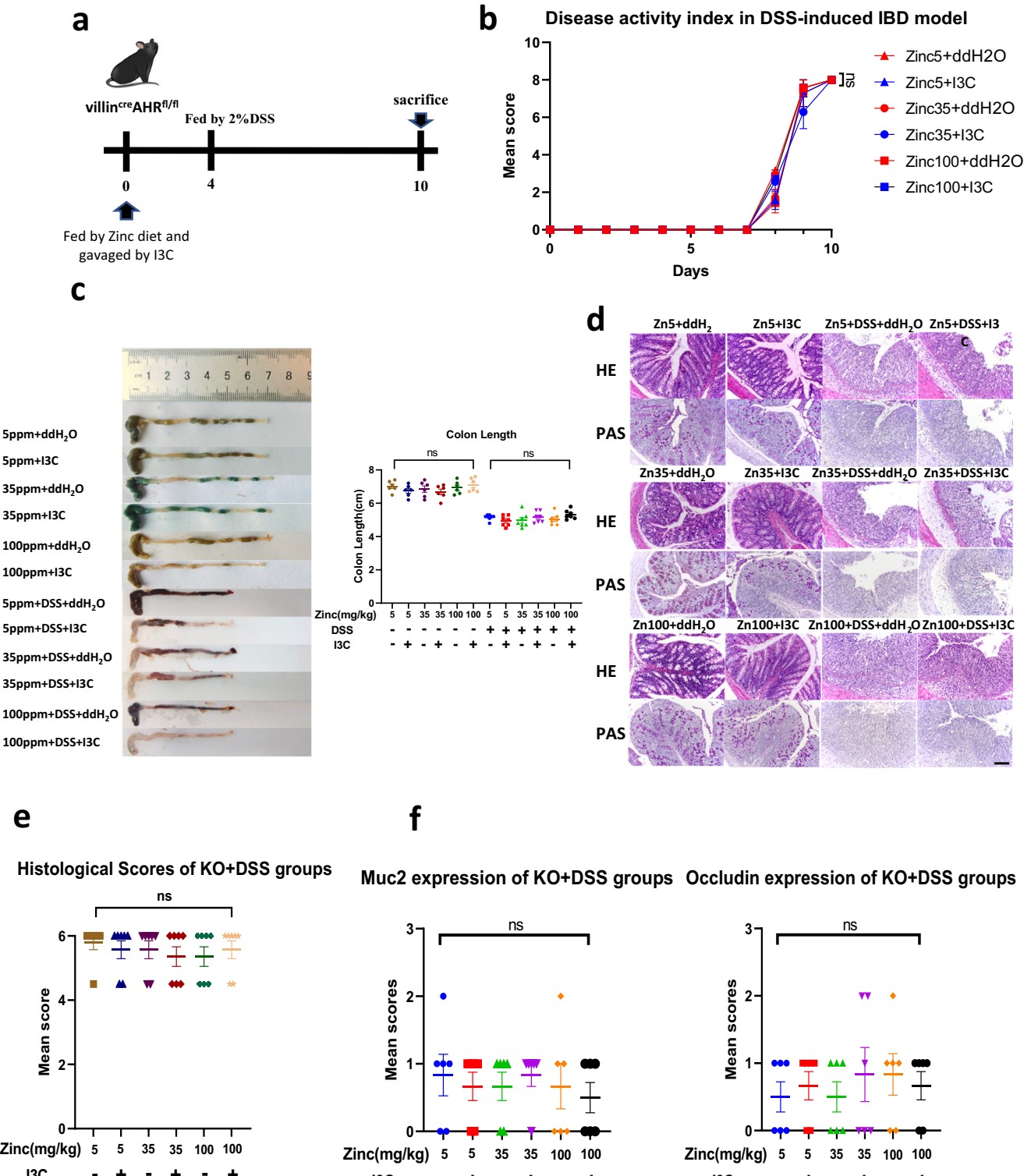

**Fig. 2 | Effects of zinc deficiency and I3C treatment on intestinal integrity in the DSS-induced IBD model in Vil1-Ahr KO mice. a** Schematic representation of the experimental schedule. Three-week-old villin^cre *Ahr*^fl/fl mice on C57BL/6 J background were provided with diets containing one of three zinc concentrations (5 mg/kg (Zinc5), 35 mg/kg (Zinc35) and 100 mg/kg (Zinc100) from Days 0 to 10 with (I3C) or without (ddH2O) I3C given by daily gavage. DSS was administered by the drinking water from Day 4 to Day10. Mice were sacrificed on Day 11. **b** Changes in intestinal disease activity index, based on diarrhoea and bleeding. Each data point represents the mean for seven animals (*n* = 7). **c** Colon image (left) and colon lengths (right) measured on Day 11. (*n* = 7 animals for DSS groups and *n* = 6 animals for control groups). **d** Histopathological changes in the colon tissue were examined by H&E and Periodic Acid Schiff staining (magnification, ×100) (repeated for 7 times). Scale bar, 100 μm. **e** Histopathological scores of the colon tissue in DSS treated groups (*n* = 7 animals). **f** Immunohistochemistry scoring of MUC2 and Occludin protein expression in colon tissue (*n* = 6 animals). Animal experiments were repeated twice. Data are presented as mean values and error bars show SEM. Statistical analysis was performed using 1-way ANOVA followed by Tukey's multiple comparison tests. ns not significant.

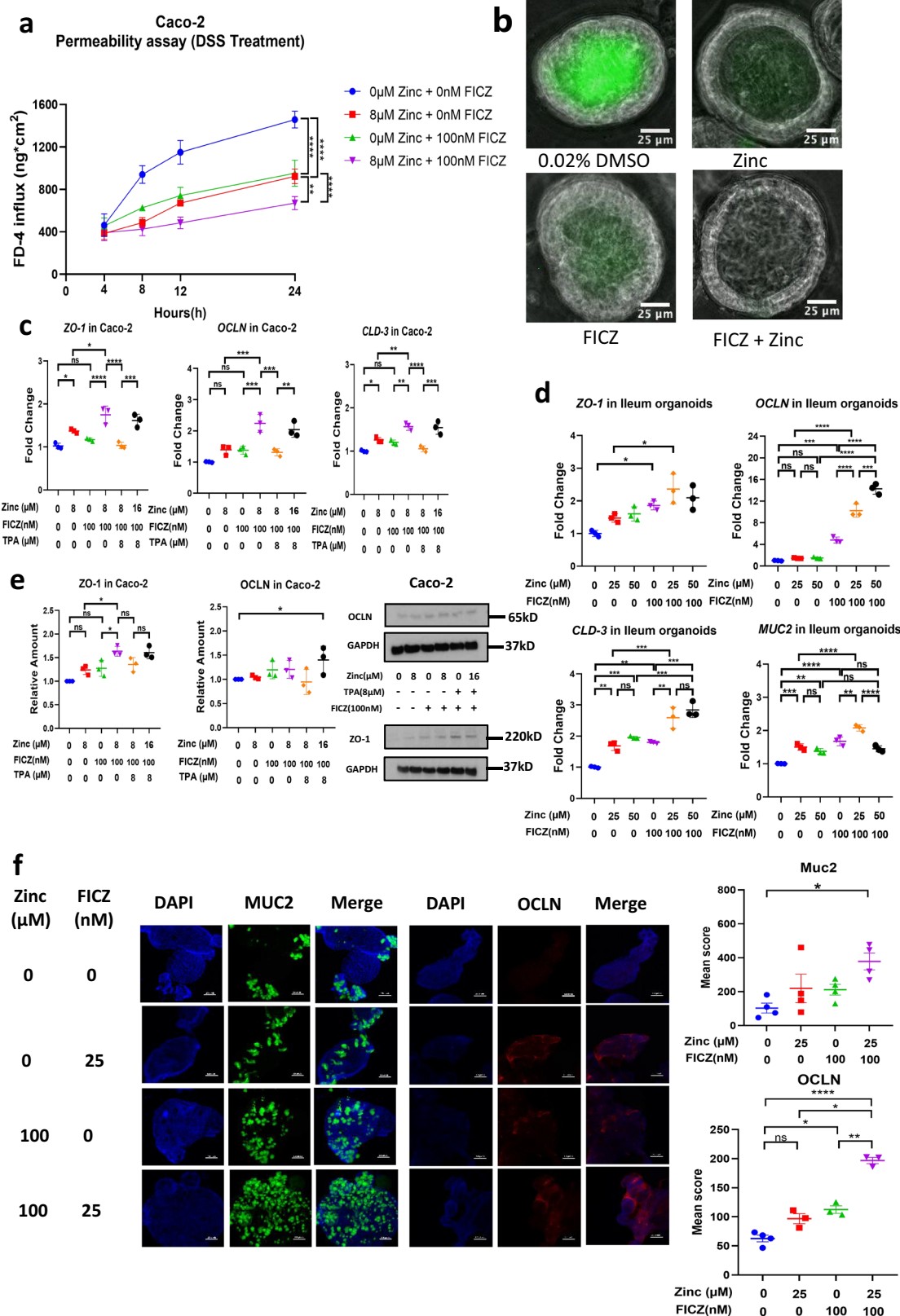

measured expression of these genes in differentiated Caco-2 cells as well as in human intestinal organoids following treatment with 10 or 100 nM FICZ. Transcripts for zinc transporters, *ZIP2, -4, -6, -7* and *-10*, which mediate zinc flux into the cytosol[40], showed increased mRNA abundance in differentiated Caco-2 cells following treatment with 10 or 100 nM FICZ, but without clear difference between the two concentrations (Fig. 5a). There was also increased expression of metal-regulatory transcription factor 1 (*MTF1*), which is an intracellular $Zn^{2+}$ sensor and transcriptional regulator of several zinc homoeostatic genes, including metallothionein 1A (*MT-1A*) and *ZIP10*. In human ileum organoids, increased gene expression was observed for *MTF1, ZIP4, -6, -7,* and *-10* (Fig. 5b) following treatment with FICZ; while in wild type

**Fig. 3 | Combined treatment of AHR agonist FICZ and zinc promotes epithelial barrier function in Caco-2 cells and human ileum organoids. a** Caco-2 cells were grown to an epithelium in a Transwell® system. At the start of the experiment, the medium in the apical compartment was replaced with MEM containing FITC-dextran 4000 (Merck) together with 3% DSS, 0 or 8 µM zinc, and 0 or 100 nM FICZ as indicated in the figure. Permeability was measured by sampling the medium in the basal compartment and measurement of FITC fluorescence over 24 h (n = 3 independent experiments). **b** human ileum organoids were challenged with 60 µM EDTA in presence of FITC-dextran in a medium containing 60 µM zinc and/or 100 nM FICZ. Leakage into the lumen was assessed after 24 h through appearance of FITC fluorescence. **c** Expression of mRNA for tight junction proteins as measured by qPCR in Caco-2 cells treated with zinc/FICZ/TPA for 24 h as indicated in the figure (n = 3 experiments). **d** Expression of mRNA for tight junction proteins and MUC2 as measured by qPCR in human ileum organoids treated with zinc and/or

FICZ for 24 h as indicated in the figure (n = 3 experiments). **e** Abundance of tight junction proteins, as measured by western blot, in Caco-2 cells treated with Zinc/FICZ/TPA for 24 h as indicated in the figure (n = 3 experiments).
**f** Immunocytochemistry (left) showing MUC2 (n = 4 experiments) and OCLN (n = 4 for control group and n = 3 for the other groups) protein levels in human ileum organoids treated with 0 or 100 nM FICZ and/0 or 25 µM zinc for 24 h, as indicated in the figure. Scale bar, 20 µm. Fluorescence levels in all regions containing DAPI and 488 nm staining were quantified using NIS Elements (Version 5.41, Nikon Instruments) and numeric data are shown (right). Statistical analysis of the data was performed using 2-way ANOVA followed by Tukey's multiple comparison tests (**a**), or 1-way ANOVA followed by Tukey's multiple comparison tests (**c–f**). Data are presented as mean values and error bars show SEM. *p < 0.05, **p < 0.01, ***p < 0.001, ****p < 0.0001, ns not significant.

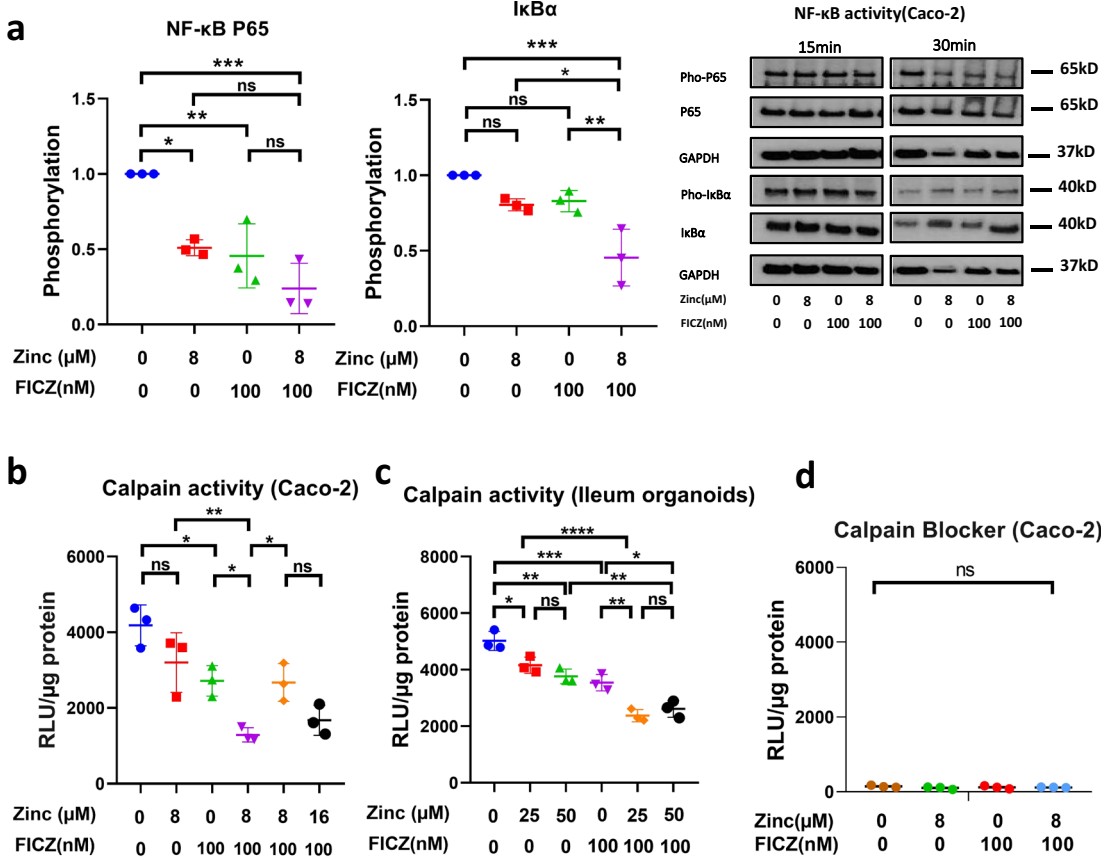

**Fig. 4 | Combined treatment of zinc and AHR agonist FICZ inhibits activities of NF-κβ and calpain. a** Caco-2 cells were treated with TNF-α to stimulate serine-536 phosphorylation of P65 and serine-32 phosphorylation of IκβΑ in media containing 0 or 8 µM zinc with or without 100 nM FICZ after 24 hours. The plots show the phosphorylation state of P65 at 30 min and IκβΑ at 15 min when maximum effect of treatments was observed. **b** Calpain activity in Caco-2 cells treated with zinc and or FICZ for 24 h. Zinc dependence of the effect was shown by addition of a zinc

chelator, TPA. **c** Calpain activity in human ileum organoids treated with different concentrations of zinc with or without FICZ for 24 h. **d** Calpain activity in Caco-2 cells after application of 50 µM calpain blocker, calpeptin. Statistical analysis of the data was performed using 1-way ANOVA followed by Tukey's multiple comparison tests. Data are presented as mean values and error bars show SEM. n = 3 experiments. *p < 0.05, **p < 0.01, ***p < 0.001, ****p < 0.0001, ns not significant.

mouse ileum organoids there was significantly increased expression of *Zip4*, *-6*, *-7*, and *-10* (Fig. 5c). We used organoids derived from ileum of villin[cre]*Ahr*[fl/fl] mice to investigate if the changes in gene expression were dependent on AHR activation. FICZ was unable to induce expression of any of the genes investigated in the AHR-deficient ileum organoids (Fig. 5d). To establish if FICZ-induced gene expression is associated with AHR recruitment to these genes, we carried out chromatin immuno-precipitation (ChIP) of AHR in FICZ-treated Caco-2 cells followed by qPCR detection of the fragments with AHR binding sites in the vicinity of the genes of interest[38,39,41]. We were able to detect

statistically significant recruitment of AHR to *ZIP4*, *-6*, *-7*, *-10* and *MTF-1*. Of these genes, *ZIP4* has a binding site for AHR within 1000 bp of the transcription start site (Supplementary Table 1) and is the most likely to be directly regulated by AHR recruitment to its promoter. ZIP4 is essential for intestinal zinc uptake, the intestinal stem cell niche, and thus for survival. Its transactivation by AHR is therefore of paramount significance. Thus, AHR regulates expression of several zinc regulatory genes in the intestinal epithelium, notably the gene for ZIP4.

To test if AHR-dependent expression of zinc importers translates to increased zinc content and cytosolic free Zn²⁺ concentration we

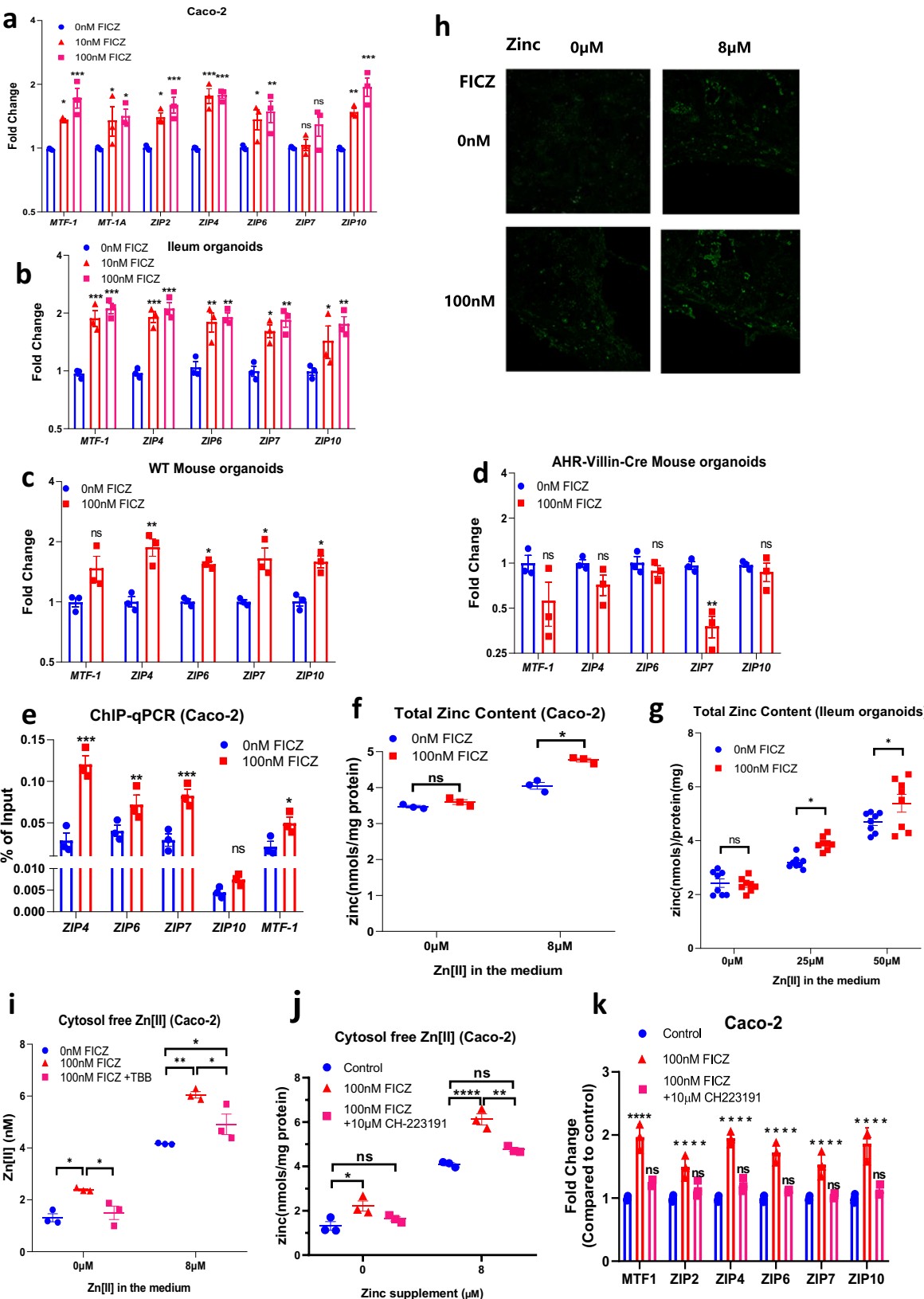

treated differentiated Caco-2 cells with 100 nM FICZ for 24 h in the absence or presence of 8 μM extracellular zinc and measured total cellular zinc content by inductively coupled plasma mass spectrometry (ICP-MS) and cytosolic free $Zn^{2+}$ concentrations by the fluorescent $Zn^{2+}$ probe, FluoZin-3AM. In the presence of 8 μM extracellular zinc, treatment with 100 nM FICZ increased the cellular zinc content,

but no change was observed when there was <0.001 μM zinc in the medium (Fig. 5f, h). Similarly, human ileum organoids treated with FICZ in the presence of different additions of extracellular zinc (0, 25, 50 μM) showed a FICZ- dependent increase in total zinc content when extracellular zinc was available (Fig. 5g). This indicates that FICZ stimulates zinc uptake in differentiated Caco-2 cells and human ileum

**Fig. 5 | AHR regulates expression of zinc importers and other zinc homoeo-static genes and stimulates cellular zinc accumulation.** Expression of zinc homoeostatic genes in **a** Caco-2 cells, **b** human ileum organoids, **c** ileum organoids from WT mice, and **d** ileum organoids from villin[cre] Ahr[fl/fl] mice after treatment with FICZ for 24 h. **e** ChIP-qPCR data showing occupancy of AHR on DNA fragments of zinc homoeostatic genes from Caco-2 cells treated with FICZ for 24 h. **f** Total zinc content in Caco-2 cells following incubation with media containing 0 or 8 μM zinc with or without 100 nM FICZ. **g** Total zinc content in human ileum organoids treated with different concentrations of zinc with or without FICZ for 24 h (n = 8 experiments). **h** Imaging of the fluorescent $Zn^{2+}$ probe, FluoZin-3AM, in Caco-2 cells after 24 h incubation in media containing 0 or 8 μM zinc with or without 100 nM FICZ. **i** Free cytosolic $Zn^{2+}$ concentrations (measured with FluoZin-3AM probe) in

Caco-2 cells following incubation with media containing 0 or 8 μM zinc with or without 100 nM FICZ and TBB. A CK2 inhibitor, TBB, was used to prevent the activating phosphorylation of ZIP7. **j** Free cytosolic $Zn^{2+}$ concentrations (measured with FluoZin-3AM probe) in Caco-2 cells following incubation with media containing 0 or 8 μM zinc with or without FICZ and CH-223191 (AHR inhibitor). **k** Expression of zinc homoeostatic genes in Caco-2 cells with or without FICZ and CH-223191. Statistical analysis of the data was performed using two-sided unpaired t tests (**e**) and 2-way ANOVA (**a**–**k**) followed by either Tukey's or Sidak's multiple comparison tests. Data are means ± SEM from three independent experiments (n = 3 per group) except for (**g**) where n = 8. *$p < 0.05$, **$p < 0.01$, ***$p < 0.001$, ****$p < 0.0001$, ns, not significant.

organoids consistent with the observed upregulation of zinc importers. Whilst the concentration of free cytosolic $Zn^{2+}$ was much lower in cells incubated for 24 h with <0.001 μM zinc in the medium compared with those kept in 8 μM zinc, treatment with FICZ resulted in an increased concentration of cytosolic free $Zn^{2+}$ regardless of the presence of zinc in the medium (Fig. 5h, i). These results indicate that FICZ treatment can also cause release of $Zn^{2+}$ into the cytosol from intracellular stores. Zinc importers ZIP2, -4, -6, and -10 can mediate cellular zinc uptake[42], but ZIP7 releases zinc from the endoplasmic reticulum (ER) into the cytosol[42]. Increased expression of *ZIP7* following FICZ treatment as reported above would explain the observation that FICZ treatment results in increased cytosolic free $Zn^{2+}$ in the absence of extracellular zinc. ZIP7 zinc transporting activity is dependent on serine phosphorylation by casein kinase-2 (CK2)[43]. To test whether FICZ increases ZIP7-mediated $Zn^{2+}$ release into the cytosol, tetra-bromobenzotriazole (TBB), a selective CK2 inhibitor, was used to reduce ZIP7 zinc transport activity, and intracellular zinc was monitored by FluoZin-3AM (Fig. 5i). TBB treatment abolished the FICZ-stimulated increase in cytosolic free $Zn^{2+}$ in the absence of extracellular zinc demonstrating that AHR activation does increase ZIP7-dependent flux of zinc from the ER into the cytosol. Also, when AHR was inhibited by the selective inhibitor CH-223191, the FICZ-stimulated increases of cytosolic free $Zn^{2+}$ (Fig. 5j) and mRNA abundance of *MTF1*, *ZIP4*, *-6*, *-7*, and *-10* (Fig. 5k) were abolished. These experiments allow us to conclude that AHR-dependent expression of ZIP zinc importers results in increased accumulation of total zinc and cytosolic free $Zn^{2+}$ in intestinal epithelial cells.

## Discussion

In this investigation, we demonstrate that AHR activation promotes barrier function by increasing tight-junction formation and mucus production by mechanisms that at least partly involve zinc uptake and increased free cytosolic $Zn^{2+}$. Specifically, we show that activation of AHR by physiologically relevant ligands induces expression of zinc importers of the ZIP family, which facilitate uptake of zinc into the epithelial cells and increases the cytosolic $Zn^{2+}$ concentration. $Zn^{2+}$ is an intracellular signalling ion which participates in a multitude of pathways, including those that regulate the development, function, and integrity of the intestinal epithelium[44–46]. Our data suggest that $Zn^{2+}$ enhances expression of genes for TJ proteins by inhibiting the NF-κß pathway, which otherwise suppresses expression of genes for the TJ proteins[47]. We also show that zinc inhibition of calpain proteolytic activity helps to maintain the levels of TJ proteins.

In this study, AHR-stimulated zinc uptake and reduced epithelial leakiness of differentiated Caco-2 cells and human ileum organoids. Treatment with I3C and zinc alleviated DSS-induced IBD in ileum and colon of mice. The protective effect of I3C was completely lost in mice that had intestinal epithelial deficiency of AHR showing that the effect was dependent on AHR specifically in these cells. Moreover, whilst in wild-type mice zinc was efficacious in alleviating IBD model symptoms, in mice lacking the AHR in the intestinal epithelium the DSS-induced lesions were equally severe in the animals receiving a zinc-

supplemented diet (100 mg/kg feed) as in those given zinc-depleted feed (5 mg/kg). Thus, AHR activity appears to be permissive of beneficial effects of zinc on the intestinal epithelium barrier function.

Caco-2 cells and ileum organoids are commonly used in vitro models to study the intestinal epithelium. Caco-2 cells lack many in vivo features, including multiple cell types and significant mucus production, while ileum organoids are more physiologically relevant but are more difficult to maintain over a long period of time. We used EDTA to challenge the permeability of human ileum organoids because it has been reported to efficiently disrupt tight junctions of ileum organoids from mice[48]. IFN-γ would have been an effective alternative[49], but it was not used in the present study as it evokes cell signalling and could interfere with the pathways interrogated. In the present study, permeability of human ileal organoids was investigated by measuring the movement of FITC-Dextran from the basolateral surface to the lumen of the organoids which is lined by the brush-border membrane. This served as a surrogate for measuring apical-to-basolateral permeability and was assumed to be a fair indication of the transepithelial leakiness. As discussed in Bardenbacher et al.[49], there are alternatives to this method including injection of dye into the lumen of the organoids or growing organoids on a Transwell system. However, such approaches are either technically demanding or require an abundance of organoid material. We consider that along with our other in vitro and in vivo evidence, reduced basolateral-to-apical permeability was sufficient to confirm the efficacy of zinc in combination with AHR activation to improve epithelial barrier function.

Both Caco-2 cells and ileum organoids have been used to study different aspects of zinc's and AHR agonists' mechanisms of action, such as gene expression, permeability, and calpain activity. By combining the two models, we were able to provide a more comprehensive understanding of the potential protective effects of zinc, AHR ligands, and their combination treatment at the cell biology level. It is worth noting that the efficacy of combined zinc and AHR agonist treatment was more pronounced in the animal model than in our in vitro systems. Whereas in wild-type mice combined treatment with I3C and zinc almost completely prevented DSS-induced injury, the effect in the Caco-2 cell injury models was partial. Both AHR and zinc have a multitude of effects on the gastrointestinal tract notably including effects on the gastroenteric immune system, which may explain the higher efficacy in vivo than in vitro[25,50,51].

The use of zinc and AHR agonists has demonstrated protective and preventative effects in the DSS-induced IBD juvenile mouse model. While these effects are promising, it is important to investigate their therapeutic potential as well. To explore the potential of zinc and AHR agonists in treatment of IBD, a chronic mouse model of IBD could be used. The therapeutic effects of zinc and AHR agonists could then be assessed by administering the treatments at different stages of the disease and monitoring their impact on disease progression and tissue regeneration. Furthermore, the chronic mouse model can also be used to investigate the potential long-term side effects of zinc and AHR agonist treatments, which is crucial for assessing their safety and efficacy as potential therapies for human IBD in the future.

Our findings conclusively show that physiological dietary AHR agonists are efficacious in preventing IBD, but only in combination with sufficient zinc intake because much of the beneficial effects of AHR are zinc-dependent. This could be critically important because approximately 1/3 of the World's population is zinc-deficient with the prevalence being highest in low-income countries where diets are dominated by plant-based foods[52]. While plant-based foods are the main source of AHR pro-ligands they also contain phytate, which makes zinc unavailable for uptake[53]. Meat and seafood are generally considered the best dietary sources of zinc, but sufficient zinc intake in the absence of these food groups could be solved with dietary supplementation or by fortifying other foods with zinc[54]. Such strategies may also be required in developed countries to combat a drift toward lower dietary zinc availability, in part driven by global sustainability issues which are prompting a move away from animal-based diets, toward plant-based foods.

A major fundamental discovery in the present study is that AHR is an important regulator of zinc fluxes and zinc-dependent processes in the intestinal epithelium. Not only does AHR regulate expression of zinc transporters, sensors and binding proteins, but in the absence of AHR the protective effect of zinc on DSS-induced IBD is completely lost (Fig. 2 and Supplementary Fig. 3). This novel finding is likely to have wider implications on other physiological and pathophysiological processes throughout the body as zinc transporters and $Zn^{2+}$ ions participate in regulation of virtually every aspect of biology including fertilisation, cell division, differentiation, growth, immunity, and cognition[55,56].

## Methods

All research reported in this study complies with all relevant regulations. Animal experiments complied with the relevant laws and institutional guidelines, as overseen by the Animal Studies Committee of the Children's Hospital of Fudan University. Consent required for the harvesting of tissue samples was given by London – Dulwich Research Ethics Committee.

### Mice

All WT and genetically modified mice used were on the C57BL/6J background. *Ahr* floxed (B6.129(FVB)-*Ahr^{tm3.1Bra/J}*) mice were purchased from the Jackson Laboratory (Bar Harbor, ME, USA). Vil1-cre mice [B6.Cg-Tg 1000 Gum] were obtained from Shanghai Model Organisms (Shanghai, China). *Ahr* floxed mice and Vil1-Cre mice were crossed for obtaining specific deletion of *Ahr* in villus epithelial cells of the small and large intestines (villin^cre*Ahr*^fl/fl) (Supplementary Fig. 9). C57BL/6J mice were purchased from SLAC laboratory animal (Shanghai, China). All mice were housed, bred, and maintained under specific pathogen-free conditions. All experiments complied with the relevant laws and institutional guidelines, as overseen by the Animal Studies Committee of the Children's Hospital of Fudan University.

### DSS-induced IBD model

In preliminary experiments, we found that the sex of mice did not affect the phenotype of DSS-induced IBD model (Supplementary Fig. 1a, b). Therefore, only one sex of mice was used in this study. For WT mouse experiments, three-week-old female C57BL/6J mice were randomly divided into 12 groups based on following treatments alone and in combinations: zinc content (5, 35, 100 mg/kg) diet, I3C or ddH2O (by oral gavage) as control, with tap water for drinking with or without 2% DSS. Zinc diets (SYSE BIO, Changzhou, China) and water with or without DSS (MP Biomedicals, OH, USA) were given ad libitum. Compositions of the experimental diets is shown in Supplementary Table 4. I3C (Sigma, Milwaukee, WI, USA) was given at a dose of 1 mg/mouse dissolved in 100 μL ddH2O by daily gavage. Zinc diets and I3C were given from the beginning to the end of the experiment and DSS was given from Day 4 to Day 11 when mice were sacrificed by

anaesthetic overdose. For the villin^cre*Ahr*^fl/fl mouse experiment, 3–4-week-old female mice were randomly divided into the same 12 groups as in the experiment with WT mice. The different zinc diets and I3C were given from beginning to end of the experiment and DSS was given from Day 4 to Day 10. Mice were sacrificed on Day 10 (not on Day 11) because of animal welfare due to serious colitis symptoms.

### Determination of disease activity index

Body weight, gross blood, and stool consistency were analysed daily. Disease activity scores were analysed according to previously published methods[57]. Briefly, for stool consistency, 0 points were assigned for well-formed pellets, 2 points for pasty and semi-formed stools that did not adhere to the anus, and 4 points for liquid stools that did adhere to the anus. For bleeding, 0 was assigned for no blood, 1 point for visible slight bleeding, and 4 points for gross bleeding.

### Histology

Colons and ileum were fixed in 10% neutral-buffered formalin. Paraffin sections were stained with haematoxylin and eosin (H&E) and Periodic Acid-Schiff[58]. Colon histological scoring was determined by the combination of scores for inflammatory cell infiltration (score 0–3) and tissue damage (score 0–3) as published previously[30]. Regarding scores for inflammatory cell infiltration, 0 points were assigned for the presence of occasional inflammatory cells in the lamina propria, 1 point for increased numbers of inflammatory cells in the lamina propria, 2 points for confluence of inflammatory cells extending into the submucosa, and 3 points for transmural extension of the infiltrate. For scoring of tissue damage, 0 points were given for no mucosal damage, 1 point for lymphoepithelial lesions, 2 points for surface mucosal erosion or focal ulceration, and 3 points for extensive mucosal damage and extension into deeper structures of the bowel wall[57].

Ileum histological scoring was determined by inflammatory cell infiltrate, epithelial changes and mucosal architecture together as follows: 0 points were assigned for normal ileum with intact epithelium and short, finger like villi, 1 point for mild mucosal inflammatory cell infiltrate, 2 points for mild diffuse inflammatory cell infiltrate in mucosa and submucosa, 3 points for moderate inflammatory cell infiltrates in mucosa and submucosa with villous blunting, 4 points for marked mucosal, submucosal and transmural inflammatory cell infiltration accompanied by villous broadening and 5 points for marked transmural inflammatory cell infiltration and villous atrophy[59].

### IHC staining and scoring

For IHC, 6 μm frozen colon sections were incubated with primary antibodies against MUC2 or Occludin, and then incubated with secondary antibodies, followed by staining and imaging. Details of the antibodies used are provided in the Supplementary Table 3. Six fields under ×400 magnification were selected from each group in a blinded way. The expression level was scored using the IHC profiler[60] plugin in ImageJ software, which allows for non-biased, automated scoring of histological slides. In this method, each DAB-stained pixel is categorised into one of four pre-set pixel intensity bins (High Positive, Positive, Low Positive, Negative). And 0 points were assigned for negative, 1 for low positive, 2 for positive and 3 for high positive.

### Microbiome profiling

The colon contents of mice were used for Microbiome profiling. Total bacterial DNA was extracted by the manufacturer's protocol (the Power Soil DNA Isolation Kit, MO BIO Laboratories). OD260/280 and OD230/260 were measured to assess the quality and concentration of extracted total DNA. The 16S rDNA genes were subjected to polymerase chain reaction amplification using the universal primer (Forward primer, GTGAATCATCGARTC; reverse primer, TCCTCCGCT TATTGAT) targeting V3 + V4 regions. High-throughput sequencing analysis of bacterial 16S rDNA genes was performed based on the

Illumina Novaseq at Biomarker Technologies Corporation (Beijing, China). According to the relationship between the overlap and paired-end reads, raw tags were obtained from the splicing sequence using Trimmomatic (version 0.33). Then, Cutadapt (version 1.9.1) was used to obtain clean tags and effective tags, respectively. Clustering of the effective tags into operational taxonomic units (OTUs) at 97% similarity was performed using USEARCH (version 10.0). Then, OTUs were annotated using RDP Classifier (version 2.2, at 0.8 confidence threshold) based on the Silva taxonomy database. Finally, alpha diversity, beta diversity and phenotype prediction were performed using QIIME2, QIIME and Bugbase, respectively.

### Generation and culture of mouse ileum organoids

mouse ileum crypts were isolated from the ileum segment of WT and villin$^{cre}$Ahr$^{fl/fl}$ mice in a PBS solution containing 2 mM EDTA[24]. The tissue was incubated for 30–45 min in a 4 °C shaking incubator (200 RPM), followed by washing and manual shaking in cold D-PBS to isolate crypts. To generate organoids, isolated crypts were embedded in Matrigel and resulting organoids were maintained in cell growth medium (WENR) supplemented with 10 μM Y-27632 (Sigma–Aldrich) was overlaid. WENR medium consists of advanced Dulbecco's modified Eagle medium (DMEM)/F12, 1× GlutaMAX, 10 mM HEPES, 100 units/mL penicillin/streptomycin, 50 ng/mL EGF, 1× B27, 1× N2 supplements (all from Gibco), 1 mM N-acetylcysteine (Sigma–Aldrich), 100 ng/mL mouse Noggin (Peprotech) and 10% R-spondin-1 conditioned medium (lab production). Three days later, the medium was changed into differentiation media[61] with no Y-27632. Organoids were passaged once a week by mechanical dissociation, at a 1:3 split ratio. Plated organoids were maintained at 37 °C in an incubator with 5% $CO_2$, and the media were changed every other day.

### Generation and culture of human terminal ileal organoids

Preparation of human terminal ileal organoids has been described before[62]. Briefly, human terminal ileum crypts were isolated from biopsies acquired from healthy individuals undergoing colonoscopy at Guy's and St. Thomas's NHS Foundation Trust with their informed consent. Biopsies were washed in cold PBS until the supernatant was clear. Following 10 min of incubation at room temperature with 10 mM 1,4-dithiothreitol (DTT), the biopsies were incubated with 8 mM EDTA in PBS and placed in a rotator for 1 h at 4 °C. At the end of the incubation, the EDTA was removed, and crypts were released with vigorous shaking in cold PBS. The crypts were further washed in PBS, pelleted and resuspended in Matrigel (Corning), in the same density as mouse crypts. human intestinal crypts embedded in Matrigel were overlaid with stem cell growth medium (WENRAS) supplemented with 10 μM Y-27632 and 5 μM CHIR99021 (Sigma–Aldrich). The human stem cell growth medium, in addition to the components of the previously described mouse medium, also contained 10 nM gastrin (Sigma–Aldrich), 500 nM A83-01 (Bio-techne), 10 μM SB202190 (Sigma–Aldrich) and 10 mM nicotinamide (Sigma–Aldrich). Three days after isolation or splitting, Y-27632 and CHIR99221 were removed from the medium, and organoids were either maintained in WENRAS or transferred into differentiation medium for setting up experiments. For the differentiation of human ileal organoids, Wnt3A surrogate protein in the medium was reduced from 0.15 nM to 0.045 nM and SB202190 and nicotinamide were withdrawn from the medium. Differentiation medium was used for 4 days, and human terminal ileal organoids were treated with different concentration of zinc with or without FICZ for 24 h. Zinc content in differentiated medium and WENRAS was measured by inductively coupled plasma mass spectroscopy (ICP-MS) and determined as 25 μM. A zinc depleted medium was created by addition of 10 μM Ca-EGTA to WENRAS. Addition of 10 μM Ca-EGTA together with 25 μM or 50 μM were considered to represent replete zinc supply and zinc supplementation, respectively.

### Cell culture

Human epithelial colorectal adherent Caco-2 cells, a gift from Prof. Paul Sharp, King's College London, were maintained in Minimum Essential Medium Eagle (MEM, Merck), supplemented with 10% fetal bovine serum (FBS, Merck), 1% 100× MEM Non-essential amino acid solution (Merck), 1% 250 μg/mL Amphotericin B (Merck), and 1% 100 units/mL penicillin (Merck), at 37 °C in a humidified atmosphere of 95% air and 5% $CO_2$. The cells were grown in the inserts of 24-well plate for more than 2 weeks at which point they differentiated into an epithelium with properties similar to enterocytes in small intestine as described previously[30].

### Whole mount staining of organoids

Organoids were isolated from the Matrigel using wash medium (advanced DMEM/F12, 10 mM HEPES and 1× GlutaMAX), and then fixed for 45 mins in 4% paraformaldehyde at room temperature. After washing using PBS-B (PBS, 0.1% BSA), organoids were permeabilized and blocked by incubation in PBS containing 3% BSA, 0.3% Triton-X-100, 1% DMSO and 5% normal goat serum (Merck) for 1 h at 20 °C and incubated overnight with primary antibodies (Supplementary Table 3). The next day, organoids were washed with PBS-B and incubated 2 h at room temperature with secondary antibodies. After washing, organoids were mounted with VECTASHIELD Vibrance Antifade Mounting Medium (Vector Laboratories). Images were captured on a Nikon A1R upright confocal microscope. We used NIS Elements (Version 5.41, Nikon Instruments) with the General Analysis 3 module license to analyse the images. A volumetric mask was generated to determine the locations in the data that are occupied by organoids. This was achieved by setting an intensity threshold from DAPI and antibody labelling. 3D binary thresholds were set for both DAPI and 488 nm channels generating 3D regions for both channels. The threshold minimum values used were 100 AU above 0 to infinity for DAPI and the same for MUC2. For immunolabelled OCLN we used a minimum of 50 AU above 0 to infinity. These were combined into a single volume which would encompass anything in the image that was an organoid. Mean fluorescence intensities from labelled antibodies targeting MUC2 or OCLN was measured within all volumes containing organoids as defined by the 3D regional mask.

### RNA extraction and quantitative real-time PCR

Total RNA of Caco-2 cells and human ileum organoids was extracted using RNAdvance Tissue kit (Beckman Coulter) and total RNA of mouse organoids was isolated using Quick-RNA Miniprep Plus Kit (Zymo Research), and all of them were assessed for purity and quantity using a Nanodrop 1000 spectrophotometer. High-Capacity RNA-to-cDNA Kit (Fisher Scientific) was used for cDNA synthesis. qPCR assays were designed using the online Universal Probe Library (UPL) assay design tool (https://lifescience.roche.com/en_gb/brands/universal-probe-library.html). Assay designs are provided in Supplementary Table 2. PCR plates were loaded using the Biomek FX liquid handling robot (Beckman Coulter) and reactions [10 ng cDNA, 0.1 μM UPL probe, 0.2 μM forward primer, 0.2 μM reverse primer and 5 μL Luna® Universal Probe qPCR Master Mix (New England Biolabs)] amplified using the Prism7900HT sequence detection system (Applied Biosystems) and analysed using sequence detection systems v2.4 software. For tissue from mice, Total RNA was extracted using Trizol Reagent. followed by reverse transcription to complementary DNA (cDNA) using the PrimeScript RT Reagent Kit (Takara, Japan). To quantify mRNA expression, cDNAs were amplified by RT-qPCR with the SYBR Premix Ex Taq RT-PCR kit (Takara, Japan) using Roche 480 Real Time PCR System. human or mouse glyceraldehyde-3-phosphate dehydrogenase (GAPDH), beta-actin (β-actin) or ubiquitin C (UBC) (Supplementary Table 3) were used as housekeeping genes. CH223191 (10 μM) was used to inhibit AHR.

## The transepithelial electricity resistance (TEER) monitor and permeability assay

Caco-2 cells ($1 \times 10^5$) were grown on 0.336 cm$^2$ Transwell inserts (Greiner Bio-One). TEER was measured with was monitored daily using an epithelial tissue volt ohmmeter (EVOMX) with STX-2 chopsticks (World Precision Instruments). TEER measurements were calculated in ohms cm$^2$ after subtracting the blank value for the membrane insert. The same inserts were employed for intestinal permeability assays. FITC-dextran 4000 (Merck), 1 mg/mL solution was put in the apical and 100 μL basement medium at 4, 8, 12, and 24 h was measured by fluorometry (excitation, 475 nm: emission, 530 nm). Serial dilutions of FITC-dextran in medium were used to determine a standard curve.

DSS was used to generate an inflammation model as described before[63]. Briefly, 2% DSS was added in the apical and basolateral compartments. Zinc and FICZ treatment, and TEER and permeability measurements were carried out as described above.

Caco-2 cell epithelia were subjected to a hypoxia challenge as published before[64]. Briefly, cells grown in 0.336 cm$^2$ Transwell inserts were transferred from a standard CO$_2$ incubator with approximately 18% O$_2$ to an atmosphere-regulated workstation set to 0% O$_2$ and 5% CO$_2$ (Sci-tive; Baker-Ruskinn, Sanford, ME, USA) for 6 h. Media were changed in the workstation to media pre-equilibrated to 0% O$_2$. After the 6-h hypoxia challenge, cells were returned to the standard CO$_2$ incubator and treated with zinc and/or FICZ. TEER and permeability measurements were the same as normal condition.

human ileum organoids in WENRAS medium, which contained 25 μM zinc, 2.9 mM calcium and 0.71 mM magnesium, were treated with 60 μM EDTA in combination with 60 μM added zinc with or without 100 nM FICZ or DMSO only for 24 h. The organoids were then washed twice with warmed PBS and incubated with 4 kDa FITC-dextran at a final concentration of 1.25 μM at room temperature for 1 h to impose a chemical serosa-to-lumen gradient. Afterwards, the FITC-dextran was removed, and the organoids were gently washed three times with PBS to remove FITC-dextran from the medium. Organoid domes were resuspended in PBS in Eppendorf tube and washed twice, before using a P1000 pipette for seeding in the Petri dish. Images were captured on a BioStation IM-Q (Nikon) and fluorescence within the organoids determined (ImageJ) by focusing on the entire luminal area of the organoid.

## Chromatin immunoprecipitation (ChIP)-qPCR

Caco-cells ($8 \times 10^6$) grown on 10 cm$^2$ dishes were fixed for 10–15 min with 1% formaldehyde, and the cross-linking reaction was stopped by addition of 1 M glycine. After cell lysis and isolation of nuclei, samples were sonicated in a Bioruptor UCD-300 in 10 mM Tris-HCl pH 8, 1 mM EDTA, 0.5 mM EGTA, 0.5% N-lauroylsarcosine to 200–500 bp fragment size. For each ChIP, 100 μg of chromatin was used, and 1/50 of chromatin as INPUT. Chromatin was immunoprecipitated with 1.1 μg of a rabbit IgG against AHR (83200 S, Cell Signaling Technology) and 1.1 μg normal rabbit IgG (2729 S, Cell Signaling Technology). Complexes were captured with Protein G Dynabeads, washed with modified RIPA buffer (50 mM HEPES pH7.5, 1 mM EDTA, 0.3% Sodium deoxycholate, 1% NP40, 250 mM LiCl), eluted in 50 mM TRIS pH8, 10 mM EDTA, 1% SDS, cross-links reversed by overnight incubation at 65 °C and DNA precipitated after phenol–chloroform extraction. AHR-target zinc regulatory genes were identified in ChIP-Atlas[65] and peaks in these genes visualised using the Integrative Genomics Viewer[66]. DNA sequences corresponding to peaks were downloaded from Ensembl (GRCh37/hg19) and interrogated for AHR elements (AHRE) by the PROMO online tool[67], which used version 8.3 of the TRANSFAC database. Unique DNA fragments were amplified and quantified by qPCR with primers directed towards AHR-binding loci containing identified AHRE in *MTF1*, *ZIP4*, *ZIP6*, *ZIP7* and *ZIP10* (Supplementary Table 1). ChIP enrichment relative to input (%) was calculated.

## Total zinc analysis

Caco-2 cells or human ileum organoids were lysed in 1 mL 20 mM Suprapur NaOH (VWR International) for 2 h. The liquid was evaporated to dryness using Genevac™ miVac Quattro Concentrator, then dissolved with 400 μL trace element grade 65–69% HNO$_3$ (Merck) and 100 μL H$_2$O$_2$ (Merck) and digested in a polypropylene tube with the lid closed overnight at 60 °C. The samples were then diluted by adding 6 mL Ultrapure 18.2 MOhms water. The zinc concentrations were determined through ICP-MS carried out using a NexIon 350 D (Perkin Elmer) at the London Metallomics Facility. The total cellular zinc content was normalised to protein content.

## Analysis of intracellular Zn$^{2+}$

After incubation with 1 μM FluoZin-3AM for 30 min at room temperature, cells were then washed and incubated in HHBSS for another 15 min at 20 °C before any imaging and readings were taken. Fluorescence was measured with 492 nm excitation and 517 nm emission in a fluorescence plate reader. Pyrithione (5 μM) together with 20 μM ZnCl$_2$ were used to treat cells for 15 min for maximum fluorescent reading, and 100 μM TPEN (N, N, N′, N′-tetrakis (2-pyridylmethyl) ethylenediamine) for 15 min as minimum fluorescence reading. Cells were preincubated with 25 μM TBB for 1 h to inhibit casein kinase 2 (CK2), which is required for ZIP7 activation. CH-223191 (10 μM) was used to inhibit AHR. Concentrations of Zn$^{2+}$ were calculated from fluorescence readings as follows, $[Zn^{2+}] = K_d (F - F_{min})/(F_{max} - F)$, using a $K_d$ of FluoZin-3am of 8.9 nM.

## BCA assay

Protein concentrations were measured by the BCA assay. Samples were diluted 1:5 in RIPA buffer, and then 1:50 with BCA assay reagent (Pierce™ BCA Protein Assay Kit, Thermo Fisher Scientific) in a 96 microtiter plate in triplicate. Absorbance was measured spectrophotometrically at 595 nm (reference: 450 nm) using a microplate reader and compared to a standard curve generated from dilutions of a known BSA standard (0.125, 0.25, 0.5, 0.75, 1.0, 1.5 and 2.0 mg/mL; Thermo Fisher Scientific).

## Protein extraction and immunoblotting

Total protein was extracted using RIPA Lysis supplement with Halt Protease Inhibitor Cocktail and Halt Phosphatase Inhibitor Cocktail (all from Thermo Fisher Scientific) following the manufacturer's protocol. Briefly, after being washed with cold PBS, cells were scraped and centrifuged for 10 min at $15,600 \times g$ at 4 °C and the protein lysate aspirated. Protein concentration was measured by BSA assay as described above and 20 μg Lysates were separated by sodium dodecyl sulfate-polyacrylamide gel electrophoresis (SDS-PAGE). Proteins were transferred onto 0.45 μm pore size nitrocellulose membranes (GE Healthcare Life Sciences) using NuPAGE™ Transfer Buffer, and membranes blocked through incubation in 5% (w/v) BSA or skim milk (1 h, RT). Membranes were incubated with primary antibodies overnight, followed by horseradish peroxidase (HRP) linked secondary antibodies. Details of the antibodies used are provided in the Supplementary Table 3. Membranes were washed three times with TBST between each step. Membranes were treated with ECL Western Blotting Detection Reagent (GE Healthcare Life Sciences) and visualised on X-ray film (GE Healthcare Life Sciences) using a film imager. Immunoblots were washed with TBST three times, incubated in stripping buffer (Thermo Fisher), and further washed with TBS-T (10 min, 20 °C). Membranes were blocked with 5% (w/v) skim milk (1 h, 20 °C) before additional antibody incubation, as described in the above. Relative abundance of proteins was determined by densiometric analysis of the X-ray films using ImageJ, followed by normalisation to GAPDH expression.

## NF-κß activity

NF-κß activity was determined indirectly by western blot and densitometric analysis of phosphorylated Iκßα$^{ser32/36}$ and P65$^{ser536}$ compared with total Iκßα and P65, respectively. The phosphorylation state of the proteins was calculated by dividing the densities of the phosphorylated protein with those of the total protein and normalising to the phosphorylation ratios of the control group.

EVP4593 (QNZ) was used as NF-κß pathway inhibitor[68] and a 4-h pre-incubation with 10 μM EVP4593 was used to block NF-κß activity.

## Calpain activity

Calpain activity was measured in Caco-2 cell or organoids extracts by Calpain-Glo™ Protease Assay (Promega) following the manufacturer's instructions. In short, cells were lysed in cell lysis buffer (Cell Signaling Technology) and centrifuged for 10 min at $15,000 \times g$ at 4 °C. The supernatant was collected for the assay. 50 μL of 50 ng/μL protein and 50 μL of the Calpain-Glo reagents were added to the luminescent calpain substrate Suc-LLVY-aminoluciferin in the presence of 1 mM $CaCl_2$. Luminescence was detected by using a BMG CLARIOstar instrument. Calpeptin has been shown to be an effective calpain blocker[69]. Pretreatment for 4 h with 50 μM calpeptin was used to block calpain activity.

## Statistical analyses

The $n$ values represent individual mice for in vivo experiments and biological replicates in cell line and organoids. Two independent experiments were repeated in animal experiments and three independent experiments were repeated in in vitro assays. Unless otherwise stated, all statistical analyses were carried out through two-sided unpaired $t$ tests, 1-way ANOVA, or 2-way ANOVA followed by either Tukey's or Sidak's multiple comparison tests, as appropriate. Data were considered statistically significant when $p < 0.05$ and presented as mean ± SEM.

## Reporting summary

Further information on research design is available in the Nature Portfolio Reporting Summary linked to this article.

# Data availability

All data that supports the findings here can be found in the manuscript and Supplementary Information. Uncropped and unprocessed scans of immunoblots have been provided as Supplementary Figures in the Supplementary Information. The 16S rDNA data generated in this study have been deposited in the NCBI Sequence Read Archive (SRA) database under accession code PRJNA945597. Source data are provided with this paper.

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

## Acknowledgements

The authors wish to thank Dr Brigitta Stockinger at The Francis Crick Institute, London, UK for valuable advice on the experiments and the

manuscript. This work was supported by grants from Guts UK (DGN2019_02 to C.H.), ZinPro Performance Minerals (1117612 to C.H.), from the National Key R&D Program of China (2021YFC2701800, 2021YFC2701802 to Y.Z.), National Natural Science Foundation of China (82241038, 81974248 to Y.Z.), Program for Outstanding Medical Academic Leader (2019LJ19 to Y.Z.), and Shanghai Committee of Science and Technology (23ZR1407600, 21140902400 to Y.Z.). X.H. was supported by a studentship from the Chinese Scholarship Council and King's College London. Metal analyses were performed in the London Metallomics Facility, which was funded by the Wellcome Trust (202902/Z/16/Z to W.M. and C.H.). Analysis of gene expression by qPCR was carried out in King's College London Genomics Centre.

## Author contributions

X.H. performed and analysed most of the experiments in cell line and organoids with input from X.L. and R.W. G.A.B. and Y.L. provided and maintained human ileum organoids, and M.R.M. provided mice ileum organoid. G.C. and A.G. performed and analysed the whole-mount image of the organoids. C.H., P.K. and W.M. conceived and designed the study with input from G.A.B., X.H. and Y.Z. W.X. performed and analysed most mice experiments with assistance from Y.G. and X.X. M.A.B. provided expertise and support for ChIP and suggestions on mouse experiments. C.H. and Y.Z. drafted the manuscript together with X.H. and W.X. All authors contributed to editing of the manuscript.

## Competing interests

The authors declare no competing interests.
