## [Peer Review file · Nature Communications]

REVIEWER COMMENTS

Reviewer #1 (Remarks to the Author):

The study by Hu and Hogstrand explores the interaction between AHR and the trace mineral, Zinc in modulating epithelial response to injury and inflammation. The study combines a DSS-induced injury model with dietary perturbations, conditional deletion approaches, in vitro mouse and human epithelial organoid cultures, and molecular biology techniques to dissect the dependence of AHR signalling on zinc in epithelial adaptation. The authors show that zinc is required for AHR-dependent protection from DSS-induced epithelial cell injury. More specifically, AHR expression in epithelial cells is key to this Zinc-dependent AHR protective function. In further experiments, the authors show that AHR activation increases total cellular and cytosolic zinc concentrations through transcriptional regulation of several SLC39 zinc importers and subsequently enhances the expression of tight junction proteins. Finally, the authors demonstrate that this modulation of TJ proteins is mediated through a zinc-dependent inhibition of NF- κ B and attenuated degradation of TJ proteins through zinc-inhibiting calpain activity. Indeed, these findings are relevant, as they explain existing clinical observations where zinc deficiency was associated with an increased risk of hospitalisations, surgeries, and disease-related complications. These outcomes, however, improve with the normalisation of zinc. The study is well performed and provides novel data, improving our understanding of how environmental and nutritional signals closely influence host-mediated adaptation in the intestine and how this could be relevant to human disease.

Minor comments:

- The manuscript was obviously formatted for another journal and needs to be reformatted to the Nat. Comm. style according to the guideline for authors: <https://www.nature.com/ncomms/submit/article>. The section headings need to be included, and also the topical subheadings of the results section need to be included.
- The authors use the DSS-induced IBD juvenile mouse model of colitis to assess the dependence of AHR ligand protection on zinc. However, the authors don't discuss the choice of this model and its advantage in comparison to using adult mice. The authors could briefly discuss this to link their findings to the observation of zinc protective function in adult IBD patients.
- The authors also used only female mice in their experiments. The authors need to explain briefly why and how they expect gender differences to be relevant to their observations.
- The material and methods do not mention the origin of the human ileal organoids. Information on whether these samples were from healthy or diseased individuals needs to be provided.
- The manuscript lack any information on the immunological status of the different mice groups used to assess whether Zinc-deficiency or AHR deletion in epithelial cells is associated with higher immune activation. Histological assessment of the infiltrating immune cells and gene expression analysis of key cytokines and chemokines from the intestinal tissue would suffice.
- Controls for the IHC detection of Muc2 and occludin are required to confirm the staining quality of the antibody used. Isotype controls are needed to evaluate the staining provided.

Reviewer #2 (Remarks to the Author):

Aryl hydrocarbon receptor utilises cellular zinc signals to maintain the gut epithelial barrier

This study by Hu et al provides evidence that the aryl hydrocarbon receptor (AHR) regulates zinc transport and absorption to main gut barrier integrity and intestinal homeostasis. The authors used cell lines, organoids, and animal models to study the role of AHR and zinc on intestinal health. They suggest that dietary supplementation with AHR ligands and zinc could be effective at preventing and treating inflammatory gut disorders.

Comments:

The role of aryl hydrocarbon receptor in intestinal health and barrier integrity is well-established with several studies publishing on the subject in recent years. Zinc is also associated with intestinal health and deficiencies in zinc are linked to IBD. Although much of the presented work is known from previous studies, the link between zinc levels and AHR provides new insight into the role that AHR

plays in barrier function. There are, however, some important concerns that the authors should address.

Suggestions for improvement:

- Given the key role of AHR in regulating cytokine levels and inflammation it was surprising that the authors did not include cytokine profiling in their in vivo analyses.
- How relevant is the oral dosing of 1 mg/mouse dissolved in 100 µl ddH₂O by daily gavage to human supplementation? The authors should also comment on why the I3C was not included in the chow together with the zinc which could better mimic human dietary exposure.
- The complete dietary composition for each of the diets used should be included in extended data.
- The in vitro studies would be improved by the inclusion of pharmacological inhibition of AHR. This is important given the interest in inhibiting AHR to improve immune mediated cancer therapy.
- Western blots should be done to confirm that the mRNA expression levels of zinc importers to confirm that the changes occur at the protein level. Pharmacological inhibition of AHR should be included on the Caco-2 and WT organoid studies to support the Ahr-villin-Cre mouse organoids.
- Western blots in Figure 4A (IkB α) are very inconsistent for GAPDH and 30 min (IkB α). These studies should be repeated.

Reviewer #3 (Remarks to the Author):

Xiuchuan Hu and colleagues investigate dietary supplements that could possibly have a benefit for patients with inflammatory bowel disease (IBD), which is characterized by compromised gut barrier function. Previous works have found that both zinc and agonists of the aryl hydrocarbon receptor may protect against IBD.

Based on experiments with a mouse model of chemically induced acute colitis, Hu et al claim that these two phenomena are linked and that the protective effect of AHR activation depends on zinc availability. These observations are partially replicated in the human colorectal cancer cell line Caco-2, a widely used in vitro model of IBD and gut barrier integrity, and human ileal organoids.

Overall, the study is well-designed and represents an important contribution to the field. To strengthen the central claim and underline the relevance in human pathology, I suggest the following revisions:

Major revisions:

1. Ileum organoids as a model of colitis:

- Please indicate whether these organoids represent a healthy gut or whether they are derived from IBD patients. Assuming that they are derived from a healthy donor, do you consider them to model colitis without chemical stimulation? A comparison to IBD patient-derived organoids and discussion of their suitability to model IBD would be very helpful in interpreting the data.
- In some experiments, i.e. FITC dextran permeability assay, the figure legend states that organoids were challenged with EDTA. Please elaborate how this was validated as a disease model (compared to DSS) and include it in your method section. Assuming EDTA is supposed to trigger a colitis-like response in organoids, the results need to be compared to organoids that were not treated with EDTA.
- The permeability assay in ileal organoids reflects diffusion from the outside into the lumen of organoids. To what extent does this reflect the physiologically more relevant direction of gut barrier penetration, i.e. from apical to basal? Consider seeding ileal organoids on transwells in analogy to Caco-2 experiments.
- Expression of tight junction genes and MUC2 is analysed by qPCR and protein levels confirmed by immunocytochemistry. The immunocytochemistry figures (3F) are unconvincing and need revision: Please show sections where DAPI staining is comparable between conditions, otherwise normalization to DAPI signal is not acceptable. Please ensure sufficient resolution of images.

2. In mouse model, knockout of AHR in gut epithelium abolishes the protective effect of Zn and AHR ligand on chemically induced colitis. Murine ileal organoids derived from AHR-KO mice reveal that AHR ligand treatment fails to upregulate TJ gene expression. Please also show effect of Zn and combined treatment in these organoids on relevant gene expression and barrier integrity (see points above). Please consider KO of AHR in human in vitro models (at least in Caco-2 cell line) to validate findings from the mouse model in a human setting.

Minor concerns:

- Figure 3F – please include a scalebar annotation and revise indication of FICZ and Zn concentration in a more comprehensible way.
- In figures where only one concentration of Zn is employed (ie figures 3F, 4A, etc), please use a uniform way of indicating FICZ and Zn concentration (either +/- or 0/25 for both).
- Please revisit colour scheme of Figure 5 to ensure accessibility to colourblind readers.
- Please distinguish between protective and therapeutic effects of Zn and AHR agonists. In many experiments a protective/preventative effect is demonstrated. Discuss how therapeutic effect could be investigated, ie in a mouse model of chronic IBD.
- Please clarify additive or synergistic effect of combined treatment (ie line 147, 154). Word choice “beneficial combination” suggests synergy, please clarify and consider quantifying combined effect.
- Please discuss the divergent features of the Caco-2 cell line and ileal organoids and how they are reflected in the obtained results.

Reviewer #4 (Remarks to the Author):

Review of manuscript NCOMMS-22-50670-T by Hu et al. for Nature Communications.

This is a convincing manuscript by HU et al. describing the important role of zinc in aryl hydrocarbon receptor (Ahr) mediated effects. This observation is new and important for understanding the function or functional loss. The authors showed in vitro and in vivo, that the activation of Ahr depends on sufficient intracellular zinc. Furthermore, they showed that the zinc-dependent activation of Ahr by the plant Ahr ligand indole-3-carbinol (I3C) reduced gut leakage. Therefore, this has a great importance for inflammatory bowel diseases. Especially, dietary zinc content as a predisposing factor may be a point to be discussed after this publication. The experiments are well performed and mostly reproducible described, see comments below. The statistics are appropriate, but the presentation should be improved. In all these data are worthy of publication and I have just some minor points, which will strengthen the manuscript.

Minor points :

1. The authors did not described the zinc compounds used in the in vivo and in vitro experiments. Since zinc bioavailability is dependent on the compound, this information is important for reproducibility and must be added.
2. There are several typos which should be corrected, e.g. line 143 “stimulate” should be “simulate”.
3. The authors used ANOVA, which is appropriate. However, it seems that they did not show all significant differences. The authors may choose to change the indication of significances with different letter, i.e. different letters indicate significant difference, as recommended by Piepho, H.-P. (2018, Letters in Mean Comparisons: What They Do and Don't Mean. *Agronomy Journal*, 110(2), 431). Otherwise the authors must include the missing significant differences, e.g. Fig.1C the difference between DSS/5mg Zn/I3C and DSS/100mg Zn/I3C. These differences are important for understanding and sometimes even discussed by the authors, but not clearly displayed in the figures.

Response to Referees Letter

Reviewer 1:

1. The manuscript was obviously formatted for another journal and needs to be reformatted to the Nat. Comm. style according to the guideline for authors: <https://www.nature.com/ncomms/submit/article>. The section headings need to be included, and also the topical subheadings of the results section need to be included.

Response: Yes, the initial submission was indeed formatted for a different journal. The manuscript has been reformatted to the standard *Nat. Comm.* style.

2. The authors use the DSS-induced IBD juvenile mouse model of colitis to assess the dependence of AHR ligand protection on zinc. However, the authors don't discuss the choice of this model and its advantage in comparison to using adult mice. The authors could briefly discuss this to link their findings to the observation of zinc protective function in adult IBD patients.

Response: Thank you for the comment. The purpose of using 3-week-old mice is to create a zinc-deficient intestinal environment more easily. It has been reported that the intestinal zinc absorption capacity of infants and children is much lower than that of adults, and infants are more vulnerable to the zinc deficiency environment¹. The zinc level in the intestines of the three-week-old mice, which had just been weaned, was more likely to be affected by the zinc content from the diet. Juvenile mice also have several advantages over adult mice in the DSS-induced IBD juvenile mouse model, including a more active intestinal epithelium and greater ability to regenerate intestinal tissue. Therefore, three-week-old mice are the best choice for animal experiments. The observed protective effects of zinc and AHR ligands in juvenile mice could be relevant to adult mice and adult IBD patients, but further studies are required to validate these findings.

3. The authors also used only female mice in their experiments. The authors need to explain briefly why and how they expect gender differences to be relevant to their observations.

Response: Thank you for the comment. In our preliminary study, the sex of mice did not affect the phenotype of DSS-induced IBD model (Fig R1). And single sex was used to reduce experimental variables (since we already have had 4 variables)

Fig R1. Weight change(left) and colon length(right) of DSS-induced IBD model in sex-matched 6-week-old mice.

A statement about this has now been included in the Methods section under subsection DSS-induced IBD model.

4. The material and methods do not mention the origin of the human ileal organoids. Information on whether these samples were from healthy or diseased individuals needs to be provided.

Response: Thank you for the comment. Human terminal ileum crypts were isolated from biopsies acquired from patients undergoing routine colonoscopy at Guy’s and St Thomas’ NHS Foundation Trust with their informed consent, as such these are healthy individuals. This information has been included under the Methods subsection “Generation and culture of human terminal ileal organoids”.

5. The manuscript lack any information on the immunological status of the different mice groups used to assess whether Zinc-deficiency or AHR deletion in epithelial cells is associated with higher immune activation. Histological assessment of the infiltrating immune cells and gene expression analysis of key cytokines and chemokines from the intestinal tissue would suffice.

Response: Thank you for this valid comment; we have now included this information. Histological assessment of the infiltrating immune cells was involved together with tissue damage in the results of colon histological scores (Supplementary Figure 1E and Supplementary Figure 3E). We also have supplemented the RT-qPCR results with expression of *Cox2*, *Il6*, *Ccl2*, *Nos2* and *Tnfa* in the colon of DSS-treated groups in the Supplementary Figure 1E and Supplementary Figure 3E.

Fig R2. RT-qPCR results of *Cox2*, *Il6*, *Ccl2*, *Nos2* and *Tnfa* in the colon from DSS-treated WT (Left) and KO (Right) groups.

6. Controls for the IHC detection of *Muc2* and *occludin* are required to confirm the staining quality of the antibody used. Isotype controls are needed to evaluate the staining provided.

Response: Thank you for this suggestion. The results of these controls have been added in the Supplementary Figure 2, which indicated the reliable staining quality of the *Muc2* and *Occludin* antibody.

Fig R3. Positive control and isotype control of Muc2 and Occludin antibody in the colon from WT mice.

Reviewer 2:

1. Given the key role of AHR in regulating cytokine levels and inflammation it was surprising that the authors did not include cytokine profiling in their in vivo analyses.

Response: Thank you for the comment. We agree and have supplemented the RT-qPCR results with data on expression of *Cox2*, *Il6*, *Ccl2*, *Nos2* and *Tnfa* in the colon of DSS-treated groups in the Supplementary Figure 1E and Supplementary Figure 3E.

Fig R2. RT-qPCR results of *Cox2*, *Il6*, *Ccl2*, *Nos2* and *Tnfa* in the colon from DSS-treated WT (Left) and KO (Right) groups.

2. How relevant is the oral dosing of 1 mg/mouse dissolved in 100 μ l ddH₂O by daily gavage to human supplementation? The authors should also comment on why the I3C was not include in the chow together with the zinc which could better mimic human dietary exposure.

Response: Thank you for the comment. According to human equivalent dose calculation based on body surface area², a 60 kg human would get about 12 times the dose of a 20g mouse. That means an adult would require about 12 mg of I3C a day to prevent IBD, which can be achieved through the daily intake of cruciferous plants or by taking I3C supplements which are readily available. Gavage has the advantage of ensuring an exact dose is delivered and circumvents possible effects of I3C on feed consumption. It is the default method used in safety studies of food supplements.

3. *The complete dietary composition for each of the diets used should be included in extended data.*

Response: Thank you for the comment. The composition for each of the diets have been added in Supplementary Table 4.

Table R1. The composition for zinc diets.

Product#	5 ppm Zinc		35 ppm Zinc		100 ppm Zinc	
	gm	kcal	gm	kcal	gm	kcal
Egg Whites, Dried	200	800	200	800	200	800
Corn Starch	150	600	150	600	150	600
Sucrose	502.38	2010	502.38	2010	502.41	2010
Cellulose, BW200	50	0	50	0	50	0
Corn Oil	50	450	50	450	50	450
Mineral MixS19401 (No Zn)	35.0	0	35.0	0	0.0	0
Mineral MixS10001	0.0	0	0.0	0	35.0	0
Zinc Carbonate, 52.1% Zinc	0.0080	0	0.0660	0	0.105	0
Vitamin MixV19401	10.4	40	10.4	40	10.4	40
Choline Bitartrate	2	0	2	0	2	0
Total	999.79	3900	999.85	3900	999.92	3900

4. *The in vitro studies would be improved by the inclusion of pharmacological inhibition of AHR. This is important given the interest in inhibiting AHR to improve immune mediated cancer therapy.*

Response: Thank you for the comment. The reason we did not include pharmacological inhibitors of AHR in our original manuscript is that efficacy and specificity of available blockers are questionable. By request, we conducted additional experiments and the attenuation of FICZ effects on cytosol free Zn(II) concentrations and expression of zinc homeostatic genes in Caco-2 cells by the AHR inhibitor CH-223191 have been added in Figure 5J and K.

Fig R4. Effect of AHR inhibitor CH-223191 on Cytosol free Zinc and expression of zinc homeostatic genes in Caco-2 cell.

5. Western blots should be done to confirm that the mRNA expression levels of zinc importers to confirm that the changes occur at the protein level. Pharmacological inhibition of AHR should be included on the Caco-2 and WT organoid studies to support the *Ahr-villin-Cre* mouse organoids.

Response: Thank you for the comment. ZIP zinc importers undergo extensive post-translational modification including sequential cleavage. For most ZIP proteins, it is therefore not possible to quantify protein abundance of the native proteins, which prohibit us from including data from WB analysis of zinc importers. mRNA expression results of pharmacological inhibition of AHR on zinc importers have been added in Figure 5K.

6. Western blots in Figure 4A (*IκBα*) are very inconsistent for GAPDH and 30 min (*IκBα*). These studies should be repeated.

Response: Thank you for the comment. The results have been done for 3 biological replicates. We agree that GAPDH may be slightly changed by differences in zinc concentrations. Considering this effect, we used the ratio of pho-*IκBα* to *IκBα*, instead of normalising pho-*IκBα* to GAPDH.

Reviewer 3:

1. Please indicate whether these organoids represent a healthy gut or whether they are derived from IBD patients. Assuming that they are derived from a healthy donor, do you consider them to model colitis without chemical stimulation? A comparison to

IBD patient-derived organoids and discussion of their suitability to model IBD would be very helpful in interpreting the data.

Response: Thank you for this interesting comment and suggestion. The organoids used in these experiments were generated from healthy individuals attending routine colonoscopy screening. Each model has its pros and cons and no model is perfect. We were not intending to use the organoids as a model of disease but as a test bed to determine the role of the AHR and zinc on maintaining barrier function. The organoids represent the most physiological *in vitro* setting for this. Disease modelling would either require use of organoids derived from patients with IBD or stimulation with a cytokine cocktail. However, modelling IBD in healthy organoids using cytokines is problematic when the focus is on barrier function. Cytokine stimulation increases inflammation markers, increases cell death, alters the polarity of some organoids but not all. Understanding the effect of AHR and zinc specifically on barrier function in this context would be challenging. Equally, IBD derived organoids are morphologically different from healthy controls, smaller with altered polarity and increased inflammation markers. The parallel experiments in Caco-2 cells were also not carried out in an IBD model for the same reason.

2. In some experiments, i.e. FITC dextran permeability assay, the figure legend states that organoids were challenged with EDTA. Please elaborate how this was validated as a disease model (compared to DSS) and include it in your method section. Assuming EDTA is supposed to trigger a colitis-like response in organoids, the results need to be compared to organoids that were not treated with EDTA.

Response: Thank you for the comment. EDTA was not used to model disease but simply to disrupt barrier function as this is the focus of our manuscript.

3. The permeability assay in ileal organoids reflects diffusion from the outside into the lumen of organoids. To what extent does this reflect the physiologically more relevant direction of gut barrier penetration, ie from apical to basal? Consider seeding ileal organoids on transwells in analogy to Caco-2 experiments.

Response: Thank you for the comment. One of the strengths of organoid technology is the apparent close resemblance to the *in vivo* architecture including a central lumen and the polarity of the cell types. This prevents easy access to the lumen which makes studying apical to basal transfer tricky. However, paracellular permeability is bi-directional and therefore permeability can be measured/assessed in either direction. There are multiple studies which have used dextran permeability to assess barrier integrity in whole organoids³⁻⁵.

4. Expression of tight junction genes and MUC2 is analysed by qPCR and protein levels confirmed by immunocytochemistry. The immunocytochemistry figures (3F) are unconvincing and need revision: Please show sections where DAPI staining is comparable between conditions, otherwise normalization to DAPI signal is not acceptable. Please ensure sufficient resolution of images.

Response: Thank you for the comment. The results have been updated in Figure 3F.

Fig R5. Immunocytochemistry (left) results of MUC2 and OCLN protein levels in human ileum organoids treated with 0 or 100 nM FICZ and/or 0 or 25 μM zinc for 24h. Fluorescence levels were quantified relative to DAPI using Fiji and numeric data are shown (Right).

5. 2a In mouse model, knockout of AHR in gut epithelium abolishes the protective effect of Zn and AHR ligand on chemically induced colitis. Murine ileal organoids derived from AHR-KO mice reveal that AHR ligand treatment fails to upregulate TJ gene expression. Please also show effect of Zn and combined treatment in these organoids on relevant gene expression and barrier integrity (see points above).

Response: Thanks for this comment, but we are not sure what these additional experiments are supposed to show that we have not already demonstrated. The key finding in our study is that the beneficial effects of dietary AHR ligands on gut barrier function are mediated by zinc through AHR stimulated induction of specific zinc importers. It was therefore relevant to include AHR KO intestinal organoids as an additional line of evidence to prove that AHR is indeed a regulator of zinc importer

expression. However, we don't feel that repeating all experiments with human ileum organoids and Caco-2 cells on mouse WT and AHR KO ileum intestine organoids is warranted.

6. 2b Please consider KO of AHR in human in vitro models (at least in Caco-2 cell line) to validate findings from the mouse model in a human setting.

Response: Thank you for the comment. We do acknowledge that KO in human in vitro models would be desirable. We attempted three times, unsuccessfully, to knock out AHR in Caco-2 cells. It is therefore unlikely that we are going to be able to follow this recommendation at this time. We hope that the additional experiments with AHR blocker will suffice to convince the reviewer of our findings.

7. Figure 3F - please include a scalebar annotation and revise indication of FICZ and Zn concentration in a more comprehensible way.

Response: Thank you for the comment. The figure has been updated.

Fig R5. Immunocytochemistry (Left) results of MUC2 and OCLN protein levels in human ileum organoids treated with 0 or 100 nM FICZ and/or 0 or 25 μM zinc for 24h. Fluorescence levels were quantified relative to DAPI using Fiji and numeric data are shown (Right).

8. In figures where only one concentration of Zn is employed (ie figures 3F, 4A, etc), please use a uniform way of indicating FICZ and Zn concentration (either +/- or 0/25 for both).

Response: Thank you for the comment. The figure has been updated and is now in line with the other figures.

Fig R6. Example (Figure 4A in the manuscript) of updated figures.

9. Please revisit colour scheme of Figure 5 to ensure accessibility to colourblind readers.

Response: Thank you for this important comment. Figure 5 in the manuscript has been updated accordingly.

Fig R7. Example (Figure 5A in the manuscript) of updated figures.

10. Please distinguish between protective and therapeutic effects of Zn and AHR agonists. In many experiments a protective/preventative effect is demonstrated. Discuss how therapeutic effect could be investigated, ie in a mouse model of chronic IBD.

Response: Thank you for the comment. Our study was intended primarily to investigate how zinc and AHR agonists combine to regulate gut epithelium permeability. Of course, this related very closely to IBD and provides medical relevance to the findings. The use of zinc and AHR agonists has demonstrated protective and preventative effects in the DSS-induced IBD juvenile mouse model. While these effects are promising, it is important to investigate their therapeutic potential as well. To investigate the therapeutic effects of zinc and AHR agonists, a chronic mouse model of IBD could be used. In this model, mice are exposed to low doses of DSS or another IBD-inducing agent over an extended period to induce chronic inflammation and tissue damage, mimicking the clinical course of human IBD. The therapeutic effects of zinc and AHR agonists can then be assessed by administering the treatments at different stages of the disease and monitoring their impact on disease progression and tissue regeneration. Furthermore, the chronic mouse model can also be used to investigate the potential long-term side effects of zinc and AHR agonist treatments, which is crucial for assessing their safety and efficacy as potential therapies for human IBD. We have now included a new paragraph in the discussion section where we propose animal experiments to investigate the potential of zinc and AHR as therapeutic treatment for IBD.

11. Please clarify additive or synergistic effect of combined treatment (ie line 147, 154). Word choice “beneficial combination ” suggests synergy, please clarify and consider quantifying combined effect.

Response: Thank you for the comment. Some of the *in vivo* results this probably do qualify as synergism in the sense that one plus the other is greater than the sum of the two, but this is not the case everywhere. Also, in pharmacology additivity is usually discussed when two drugs (or other xenobiotics) act on the same protein, but in this case AHR and zinc may act on multiple targets and AHR activation is permissive of the effects caused by zinc. The picture is further complicated by the combination of single doses of AHR agonists with several zinc doses. We therefore agree that we cannot state from our data whether there is interaction in a pharmacological sense and should avoid implying additivity or synergism. Thus, synergy and have replaced ‘beneficial effect’ with ‘benefit’ or ‘beneficial combination’ in the manuscript.

12. Please discuss the divergent features of the Caco-2 cell line and ileal organoids and how they are reflected in the obtained results.

Response: Thank you for the comment. Caco-2 cells and ileal organoids are commonly used *in vitro* models to study the intestinal epithelium. Caco-2 cells lack many features of the intestinal epithelium, *in vivo*, while ileal organoids are more physiologically relevant but can be challenging to maintain over a long period of time. Both models have been used to study different aspects of zinc and AHR agonists' mechanism of action, such as gene expression, permeability, and calpain activity. To strengthen our results, we verified most experiments conducted in Caco-2 cells in ileal organoids. By combining the two models, we were able to provide a more comprehensive understanding of the potential therapeutic effects of zinc, AHR ligands, and their combination treatment at the cell biology level. We have added some more details about the two *in vitro* systems where they are introduced in the results section and a discussion about their divergent and complementary features in the second paragraph of the 'Discussion' section.

Reviewer 4

1. The authors did not described the zinc compounds used in the in vivo and in vitro experiments. Since zinc bioavailability is dependent on the compound, this information is important for reproducibility and must be added.

Response: Thank you for the comment. Zinc carbonate was used in the *in vivo* experiments and zinc sulfate was used in the *in vitro* experiments. The composition for each of the diets have been added in the supplementary Table 4. The choice of zinc supplement for *in vivo* and *in vitro* experiments may depend on different factors such as stability, solubility, bioavailability, and taste or palatability. Zinc carbonate (ZnCO₃) is a good choice for *in vivo* experiments because of its stability and neutral taste, which can provide a consistent and reliable dose of zinc over a long-term study without altering palatability of animal feed or drinking water. On the other hand, zinc sulfate (ZnSO₄) is a good choice for *in vitro* experiments because of its high solubility, which can provide a consistent and reliable source of zinc to cells in culture. Since cells were treated with ZnSO₄ for only 24 hours in most *in vitro* experiments, a highly bioavailable form of zinc was needed. ZnSO₄ is more readily absorbed by cells and can be efficiently incorporated into biological processes compared to ZnCO₃, making it the optimal choice for our *in vitro* experiments.

Table R1. The composition for zinc diets.

Product#	5 ppm Zinc		35 ppm Zinc		100 ppm Zinc	
Ingredient	gm	kcal	gm	kcal	gm	kcal

Egg Whites, Dried	200	800	200	800	200	800
Corn Starch	150	600	150	600	150	600
Sucrose	502.38	2010	502.38	2010	502.41	2010
Cellulose, BW200	50	0	50	0	50	0
Corn Oil	50	450	50	450	50	450
Mineral Mix S19401 (No Zn)	35.0	0	35.0	0	0.0	0
Mineral Mix S10001	0.0	0	0.0	0	35.0	0
Zinc Carbonate, 52.1% Zinc	0.0080	0	0.0660	0	0.105	0
Vitamin Mix V19401	10.4	40	10.4	40	10.4	40
Choline Bitartrate	2	0	2	0	2	0
Total	999.79	3900	999.85	3900	999.92	3900

2. The authors used ANOVA, which is appropriate. However, it seems that they did not show all significant differences. The authors may choose to change the indication of significances with different letter, i.e. different letters indicate significant difference, as recommended by Piepho, H.-P. (2018, Letters in Mean Comparisons: What They Do and Don't Mean. *Agronomy Journal*, 110(2), 431). Otherwise the authors must include the missing significant differences, e.g. Fig. 1C the difference between DSS/5mg Zn/I3C and DSS/100mg Zn/I3C. These differences are important for understanding and sometimes even discussed by the authors, but not clearly displayed in the figures.

Response: Thank you for the comment. The figure has been updated. We attempted different approach to denote significance, included the one the Referee kindly refers to. Because of the multiple conditions and several control groups, showing all significant differences rendered the figures very complicated. We therefore provide the complete statistical analysis in the source data file accompanying the manuscript.

Table R2. The ANOVA analysis of Figure. 1C

Tukey's multiple comparisons test	Mean Diff.	95.00% CI of diff.	Summary	Adjusted P
Zn5+DSS+ddH2O vs. Zn5+DSS+I3C	-0.2833	-0.9541 to 0.3874	ns	0.7907
Zn5+DSS+ddH2O vs. Zn35+DSS+ddH2O	-0.7333	-1.404 to -0.06260	*	0.0257
Zn5+DSS+ddH2O vs. Zn35+DSS+I3C	-1.683	-2.354 to -1.013	****	<0.0001
Zn5+DSS+ddH2O vs. Zn100+DSS+ddH2O	-0.7667	-1.437 to -0.09593	*	0.0178
Zn5+DSS+ddH2O vs. Zn100+DSS+I3C	-1.633	-2.304 to -0.9626	****	<0.0001
Zn5+DSS+I3C vs. Zn35+DSS+ddH2O	-0.45	-1.121 to 0.2207	ns	0.3444
Zn5+DSS+I3C vs. Zn35+DSS+I3C	-1.4	-2.071 to -0.7293	****	<0.0001
Zn5+DSS+I3C vs. Zn100+DSS+ddH2O	-0.4833	-1.154 to 0.1874	ns	0.2711

Zn5+DSS+I3C vs. Zn100+DSS+I3C	-1.35	-2.021 to -0.6793	****	<0.0001
Zn35+DSS+ddH2O vs. Zn35+DSS+I3C	-0.95	-1.621 to -0.2793	**	0.0021
Zn35+DSS+ddH2O vs. Zn100+DSS+ddH2O	-0.03333	-0.7041 to 0.6374	ns	>0.9999
Zn35+DSS+ddH2O vs. Zn100+DSS+I3C	-0.9	-1.571 to -0.2293	**	0.0038
Zn35+DSS+I3C vs. Zn100+DSS+ddH2O	0.9167	0.2459 to 1.587	**	0.0031
Zn35+DSS+I3C vs. Zn100+DSS+I3C	0.05	-0.6207 to 0.7207	ns	>0.9999
Zn100+DSS+ddH2O vs. Zn100+DSS+I3C	-0.8667	-1.537 to -0.1959	**	0.0056

References

1. Krebs, N. F., Miller, L. V & Hambidge, K. M. Zinc deficiency in infants and children: a review of its complex and synergistic interactions. *Paediatr. Int. Child Health* **34**, 279–288 (2014).
2. Reagan-Shaw, S., Nihal, M. & Ahmad, N. Dose translation from animal to human studies revisited. *FASEB J. Off. Publ. Fed. Am. Soc. Exp. Biol.* **22**, 659–661 (2008).
3. Bardenbacher, M. *et al.* Investigating Intestinal Barrier Breakdown in Living Organoids. *J. Vis. Exp.* (2020). doi:10.3791/60546
4. Pearce, S. C. *et al.* Marked differences in tight junction composition and macromolecular permeability among different intestinal cell types. *BMC Biol.* **16**, 19 (2018).
5. Li, B. *et al.* Intestinal epithelial tight junctions and permeability can be rescued through the regulation of endoplasmic reticulum stress by amniotic fluid stem cells during necrotizing enterocolitis. *FASEB J. Off. Publ. Fed. Am. Soc. Exp. Biol.* **35**, e21265 (2021).

REVIEWER COMMENTS

Reviewer #1 (Remarks to the Author):

I thank the authors for responding to all my questions and remarks.

Reviewer #2 (Remarks to the Author):

This study provides convincing evidence that the aryl hydrocarbon receptor (AHR) regulates zinc transport and absorption to regulate barrier integrity and intestinal homeostasis. In this revised version, the authors appropriately addressed comments raised during the initial review. Overall, the study is well designed, and the methodology is sound. The conclusions are supported by comprehensive and solid data. This important work provides new insight into the significance and complexity of AHR's role in intestinal health.

Reviewer #3 (Remarks to the Author):

The authors have revised their manuscript to address a number of referee requests. Nevertheless, several crucial points from the previous review round were dismissed with minimal effort in the rebuttal and revised manuscript and require more attention to clarify aspects of methodology and findings from this paper.

R3.2: Thank you for the clarification of the usage of EDTA. Considering that EDTA is a zinc-chelating agent and it might interfere with the Zinc treatment, did you consider another way of inducing barrier disruption, ie IFN γ as done previously (e.g. PMID: 30776676)? Comparing to unchallenged organoids would be useful to show the baseline barrier integrity. The Zn and FICZ concentrations used for the organoid experiments could not be found easily in the manuscript, please include in method section and figure legend.

R3.3: Indeed studying apical-to-basal transport is more tricky in organoids. It could be achieved through a number of ways, i.e. microinjection of FITC-dextran into the lumen and measuring the loss of luminal fluorescence, or using inside-out intestinal organoids, or seeding organoids as monolayers on transwells. I think that the method described here is sufficient to show increased leakiness for passive diffusion considering the supporting evidence from other models, but please consider including a statement similar as in e.g. PMID: 30776676 to highlight that the direction of permeability is opposite to the in vivo situation.

R3.4: The data shown in the quantification looks identical to the data shown in the first version of the manuscript, with the exception that for MUC2 condition 25 μ M Zn/noFICZ there were 4 data points before, now only 3. The figure legend indicates data from 3 independent experiments, but it is unclear how many organoids were assessed per experiment. Please clarify how many organoids and images are underlying the quantification and how divergent DAPI levels are accounted for (inclusion/exclusion criteria). Considering the divergent DAPI levels and organoid morphologies, several organoids per experiment should be quantified to allow robust statements.

R3.5: The main claim of the paper that the beneficial effects of dietary AHR ligands on gut barrier function are mediated by zinc, would be stronger if you could assess barrier integrity effects of Zn in wt and AHR KO mouse organoids in response AHR treatment. In this case, the in vitro experiments would provide crucial support for the circumstantial in vivo data on barrier integrity-related proteins, but where permeability has not been measured. Considering that these organoids and assays are in place, I think this experiment should be feasible and insightful for actual barrier integrity impact in this

setting. Related to this, a statement on validation of AHR KO in the mouse tissues or organoids by qPCR and WB should be included in the manuscript.

R3.7: At the bottom right picture of 3F, the scale bar seems thick and vertical, please consider revising. Please indicate size of scale bar in the figure legend, i.e. "scalebar indicates xxx μm ", also for figure 3B and any other figures with microscope images.

R3.10: The authors consistently refer to the observed effects as "therapeutic" throughout the manuscript but do add a disclaimer in the discussion that other experimental setups, e.g. in long-term IBD mouse models, may be necessary to prove therapeutic effects over the preventative and protective effects shown here. Please consider revising.

Reviewer #4 (Remarks to the Author):

The authors responded all comments of the reviewers adequately.

Where the authors denied to do additional experiments, the rational is given and the argumentation of the authors is clear and founded on their as well as data already published. Therefore, these additional experiments are not needed and fast publication is more helpful for the scientific community than additional data without further conclusion from the dataset.

However, the authors did not include all figures, e.g. FigR1, of the rebuttal in the supplementary data. It is nice to convince the reviewers, but it is not fair to restrain this information from the reader. Therefore I strongly encourage the authors to include all information from the rebuttal into the supplementary material, to present the whole picture of the experiments done.

Revision 2

Responses to Referees' Comments

Reviewers 1 and 2:

No further comments

Response: We thank the reviewers for their constructive comments which have substantially improved the manuscript.

Reviewer 3:

R3.2: Thank you for the clarification of the usage of EDTA. Considering that EDTA is a zinc-chelating agent and it might interfere with the Zinc treatment, did you consider another way of inducing barrier disruption, ie IFN γ as done previously (e.g. PMID: 30776676)? Comparing to unchallenged organoids would be useful to show the baseline barrier integrity. The Zn and FICZ concentrations used for the organoid experiments could not be found easily in the manuscript, please include in method section and figure legend.

Response: This is a valid point, which we considered during the design of our study. EDTA binds zinc, calcium or magnesium with 1:1 stoichiometry. The added 60 μ M EDTA would have chelated some of the 60 μ M zinc added to the WENRAS medium, but calcium and magnesium were present at a much higher concentrations ([Ca] = 2.9 mM; [Mg] = 0.71 mM) and the medium before zinc addition contained 25 μ M zinc (see Methods section). Therefore, EDTA and zinc both had their intended effects on epithelial permeability.

In planning the study, we discussed different experimental paradigms to disrupt the epithelial barrier of the organoids, including the use of IFN- γ as elegantly demonstrated in the Bardenbacher et al. 2019 paper, which Reviewer 3 refers to. Our concern was that an inflammatory cytokine, such as IFN- γ , could interfere with the cell signalling pathways evoked by zinc and FICZ treatment and therefore confound the results. Therefore, we preferred the more physical approach of tight junction disruption using EDTA. As this worked, we did not consider it necessary to add further experimental paradigms.

We have added further details about the experimental protocol including zinc and FICZ concentrations in the Methods section and legends for Figure 3 and Supplementary Figure 5. We also added a brief discussion on pages 15 and 16 of the manuscript about

Fig R1. Baseline barrier integrity of the human ileum organoids compared with the effect of EDTA.

the alternative to induce barrier disruption with IFN- γ citing Bardenbacher et al. (2019).

A new panel has been added to Supplementary Figure 5 and shown here (Fig R1) showing baseline barrier integrity of the human ileum organoids compared with the effect of EGTA without addition of zinc or FICZ.

R3.3: Indeed studying apical-to-basal transport is more tricky in organoids. It could be achieved through a number of ways, i.e. microinjection of FITC-dextran into the lumen and measuring the loss of luminal fluorescence, or using inside-out intestinal organoids, or seeding organoids as monolayers on transwells. I think that the method described here is sufficient to show increased leakiness for passive diffusion considering the supporting evidence from other models, but please consider including a statement similar as in e.g. PMID: 30776676 to highlight that the direction of permeability is opposite to the in vivo situation.

Response:

We followed the recommendation by Reviewer 3 and introduced a discussion on page 16 about the limitation of our experimental approach and alternative methods that could be used, including the limitations of these.

R3.4: The data shown in the quantification looks identical to the data shown in the first version of the manuscript, with the exception that for MUC2 condition 25 μ M Zn/noFICZ there were 4 data points before, now only 3. The figure legend indicates data from 3 independent experiments, but it is unclear how many organoids were assessed per experiment. Please clarify how many organoids and images are underlying the quantification and how divergent DAPI levels are accounted for (inclusion/exclusion criteria). Considering the divergent DAPI levels and organoid morphologies, several organoids per experiment should be quantified to allow robust statements.

Response:

We have re-introduced the fourth data-point in Figure 3F for the 25 μ M Zn, 0 FICZ condition. There was a small error in the annotation of the statistics of Figure 3F in the initial submission. This has now been corrected and it does not change the outcome or alter the conclusions.

We thank the referee for drawing our attention to an error in the legend for Figure 3. Fluorescence signals were not normalised to DAPI in the quantitative analysis of the data as presented. This has now been corrected. Instead DAPI staining in combination with fluorescence from immunolabelling of MUC2 and OCLN was used to generate a

3D mask of all regions occupied by organoids (Fig R2). Mean fluorescence intensities from labelled antibodies targeting MUC2 or OCLN were measured within all volumes containing organoids as defined by the 3D regional mask. Therefore, fluorescence data presented are from all organoids in each replicate of the experiment. Since DAPI signal was not used for normalization, variation in the DAPI signal did not influence the data presented, which is why the data shown in Fig 3F has not changed. Organoid differed in size and numbers in each replicate, but we did not count the number of organoids we took readings from because we measured fluorescence in all volumes that contained organoids in each replicate. We have extended the Methods section with a description about this workflow, including fluorescence thresholds.

Fig R2. Mask of regions occupied by organoids generated by capturing fluorescence from DAPI and immunostaining. Mean fluorescence intensities from labelled antibodies targeting MUC2 or OCLN were measured within all volumes containing organoids as defined by the 3D regional mask.

R3.5: The main claim of the paper that the beneficial effects of dietary AHR ligands on gut barrier function are mediated by zinc, would be stronger if you could assess barrier integrity effects of Zn in wt and AHR KO mouse organoids in response AHR treatment. In this case, the in vitro experiments would provide crucial support for the circumstantial in vivo data on barrier integrity-related proteins, but where permeability has not been measured. Considering that these organoids and assays are in place, I think this experiment should be feasible and insightful for actual barrier integrity impact in this setting. Related to this, a statement on validation of AHR KO in the mouse tissues or organoids by qPCR and WB should be included in the manuscript.

Response:

We have added a statement about validation of AHR KO in intestinal epithelium of mice used in the animal experiment and an additional figure (Supplementary Fig. 9; Fig R3) showing results from genotyping and western blot for AHR. Mouse intestinal organoids were provided by our Crick Institute collaborator and the AHR KO strain has been described before (Metidji et al., 2018, Immunity, doi.org/10.1016/j.immuni.2018.07.010).

Fig R3. Verification of intestinal epithelium deletion of Ahr.

Ahr floxed mice and Vil1-Cre mice were crossed for obtaining specific deletion of Ahr in villus epithelial cells of the small and large intestines (villincreAhrfl/fl). (A). Genotyping results of AhR loxp (+/+) Vil1-cre (+/-) mice. (B). Western Blotting results of AhR in intestinal epithelial cells from C57BL/6J and AhR loxp (+/+) Vil1-cre (+/-) mice. (C). Western Blotting results of AhR in liver tissue from C57BL/6J and AhR loxp (+/+) Vil1-cre (+/-) mice.

We agree that data on barrier integrity effects of Zn in WT vs. AHR KO mouse organoids in response to FICZ would be nice to have, but not that these would be required to demonstrate that AHR has zinc-mediated effects that are beneficial for the intestinal barrier function. We refer to Reviewers 1, 2 and 4 who conclude that our claims are fully supported by the data presented. Our main lines of evidence that zinc mediates beneficial effects of AHR activation on the intestinal epithelium barrier function are as follows:

- I3C offers no protection against IBD in zinc deficient WT mice and is enhanced by zinc supplementation. Hence, I3C protection needs zinc.
- I3C offers no protection against IBD at any zinc dose in AHR KO mice. Hence, I3C protection is mediated by AHR.
- I3C/FICZ increases expression of genes and proteins contributing to barrier function (TJ proteins and mucin-2) in mouse intestine, human ileum organoids, and Caco-2 cells in an AHR and zinc dependent manner.
- FICZ reduces epithelial permeability of human ileum organoids and Caco-2 cells in a zinc-dependent manner.

- FICZ upregulates expression of zinc importers (Caco-2, human organoids, mouse organoids) but not in mouse AHR KO organoids or after blocking AHR in Caco-2 cells.
- Zinc uptake in Caco-2 cells and human ileum organoids is enhanced by FICZ treatment but not after blocking AHR.
- AHR mediates upregulation of TJ genes through zinc inhibition of NF- κ B and prevents TJ protein degradation through zinc inhibition of calpain.

We consider these lines of evidence conclusively showing that AHR activation stimulates zinc uptake which promotes tight junction formation, increases mucous production, and protects against IBD. This does not preclude other effects that AHR and zinc may have individually on the epithelium as already stated in the manuscript. An additional mechanism published after the submission of our manuscript involves epigenetic changes in expression of claudins brought about via basolateral influx of zinc into the intestinal epithelium via zinc transporter Zip14 (Jimenez-Rondan et al., 2023, <https://doi.org/10.1152/ajpgi.00244.2022>) and more are likely to follow. Furthermore, considering the imperfect concordance of treatment related effects between rodents and human, the proposed experiments would be better carried out with human KO organoids, which we do not yet have. In conclusion, the additional experiments proposed would be nice, but would not add to the conclusions. Therefore, we respectfully suggest that a prompt publication of this study is more important than adding these additional experiments, which should be carried out in human AHR KO organoids generated for future studies.

R3.7: At the bottom right picture of 3F, the scale bar seems thick and vertical, please consider revising. Please indicate size of scale bar in the figure legend, i.e. “scalebar indicates xxx μ m”, also for figure 3B and any other figures with microscope images.

Response:

We have fixed the issue with the scalebars in Figure 3F and added in the legend as requested: “Scalebar, 20 μ m.” The size of the scalebar in Figure 3B is shown in the figure.

R3.10: The authors consistently refer to the observed effects as “therapeutic” throughout the manuscript but do add a disclaimer in the discussion that other experimental setups, e.g. in long-term IBD mouse models, may be necessary to prove therapeutic effects over the preventative and protective effects shown here. Please consider revising.

Response:

We have replaced the word “therapeutic” everywhere except in the abstract where we speculate that treatment with zinc and AHR agonist might also be therapeutic.

Reviewer 4

1. The authors responded all comments of the reviewers adequately.

Where the authors denied to do additional experiments, the rational is given and the argumentation of the authors is clear and founded on their as well as data already published. Therefore, these additional experiments are not needed and fast publication is more helpful for the scientific community than additional data without further conclusion from the dataset.

Response: We thank the reviewer for their constructive comments which have substantially improved the manuscript. We agree that conclusions are supported by the data and much appreciate the proposal to move forward with the publication.

2. However, the authors did not include all figures, e.g. FigR1, of the rebuttal in the supplementary data. It is nice to convince the reviewers, but it is not fair to restrain this information from the reader.

Therefore, I strongly encourage the authors to include all information from the rebuttal into the supplementary material, to present the whole picture of the experiments done.

Option 1

Response: Thank you for the comment. We have added the figure demonstrating no difference in response to DSS treatment in males and females as supplementary Figure 1.